# Architectural and Inferential Inductive Biases for Exchangeable Sequence Modeling

**Daksh Mittal**,* **Ang Li**,* **Thomson Yen**,* **Daniel Guetta**, **Hongseok Namkoong**
Columbia University
{dm3766, al4263, ty2531, crg2133, hn2369}@columbia.edu

## Abstract

Autoregressive models have emerged as a powerful framework for modeling exchangeable sequences—i.i.d. observations when conditioned on some latent factor—enabling direct modeling of uncertainty from missing data (rather than a latent). Motivated by the critical role posterior inference plays as a subroutine in decision-making (e.g., active learning, bandits), we study the inferential and architectural inductive biases that are most effective for exchangeable sequence modeling. For the inference stage, we highlight a fundamental limitation of the prevalent single-step generation approach: its inability to distinguish between epistemic and aleatoric uncertainty. Instead, a long line of works in Bayesian statistics advocates for multi-step autoregressive generation; we demonstrate this "correct approach" enables superior uncertainty quantification that translates into better performance on downstream decision-making tasks. This naturally leads to the next question: which architectures are best suited for multi-step inference? We identify a subtle yet important gap between recently proposed Transformer architectures for exchangeable sequences Müller et al. [22], Nguyen and Grover [23], Ye and Namkoong [30], and prove that they in fact cannot guarantee exchangeability despite introducing significant computational overhead. Through empirical evaluation, we find that these custom architectures can significantly underperform compared to standard causal masking, highlighting the need for new architectural innovations in Transformer-based modeling of exchangeable sequences.

## 1 Introduction

Intelligent agents must be able to articulate their own uncertainty about the underlying environment, and sharpen its beliefs as it gathers more information. However, uncertainty quantification can be difficult in general without a complete characterization of the nature of the data and sources of uncertainties. *Exchangeable sequences* represent an important class of data structures on which one can quantify both the nature and degree of uncertainty principally, thereby enabling more robust decision-making algorithms. To characterize this concretely, consider a sequence of observations $Y_{1:\infty}$ gathered from an unseen environment $\theta$, e.g., noisy answers in a math quiz, generated by a student's current proficiency level $\theta$. When marginalized over the latent $\theta$, the joint distribution of the sequence $Y_{1:\infty}$ is permutation invariant. This property defines what are known as *exchangeable sequences*, which serve as a fundamental unit of study in uncertainty quantification of latent variables that govern data generation.

Autoregressive sequence modeling has recently gained significant attention as a powerful approach for modeling exchangeable sequences $Y_{1:\infty}$ [22, 23, 31, 30, 20]. Unlike conventional Bayesian modeling—which requires specifying a prior over an unobserved latent variable $\theta$ and a likelihood for the observed data, often a challenging task—autoregressive sequence modeling builds on De Finneti's

---

*Equal contribution

39th Conference on Neural Information Processing Systems (NeurIPS 2025).

predictive view of Bayesian inference [6, 7, 8, 9]. This view directly models the observables $Y_{1:\infty}$: for an exchangeable sequence, epistemic uncertainty in the latent variable $\theta$ stems from the unobserved future data [2, 4, 12]. Viewing future observations as the sole source of epistemic uncertainty in $\theta$ for exchangeable sequences [2, 4, 12], autoregressive sequence modeling enables direct prediction of the observables $Y_{1:\infty}$, offering a conceptually elegant and practical alternative to conventional Bayesian approaches.

Transformers have emerged as the dominant architecture for autoregressive sequence modeling [20, 22, 17, 18, 15, 29], owing to their remarkable performance in natural language and vision applications [5, 10]. As Transformers are increasingly employed to meta-learn probabilistic models for large-scale tabular datasets [32, 20], they offer a unique opportunity to move beyond traditional prediction tasks or merely replicating supervised algorithms—an area that has been the primary focus so far. Instead, following De Finetti's perspective, when meta-trained on tabular datasets, these models can effectively quantify epistemic uncertainty, which powers decision-making and active exploration across diverse domains, including recommendation systems, adaptive experimentation, and active learning [31]. For instance, we can train sequence models on a collection of tables, each representing a different disease diagnosis setting—akin to applying meta-learning in the context of disease diagnosis. These models can then be leveraged to actively gather additional data in a previously unseen disease diagnosis setting to enhance model's predictive performance (see Figure 1 for illustration).

However, using Transformers to model exchangeable sequences for decision-making presents its own challenges. Since the existing literature has primarily focused on traditional prediction tasks or the replication of supervised algorithms rather than decision-making, it has overlooked the perspective that epistemic (reducible) uncertainty stems from missing data. Accurately distinguishing between epistemic (reducible) and aleatoric (irreducible) uncertainty is crucial for decision-oriented applications. A significant limitation in the current literature is its predominant focus on one-step predictive uncertainty [22, 23] or one-step predictions [18], which fails to differentiate between epistemic and aleatoric uncertainty. In contrast, a long line of work in Bayesian statistics [28, 24] advocates multistep inference, offering a more robust framework for differentiating between epistemic and aleatoric uncertainty.

Our **first key contribution** is to empirically and theoretically demonstrate the limitations of one-step inference in sequence models for uncertainty quantification (Section 3). Specifically, we show that one-step inference leads to a loss of information (Theorem 2), which degrades uncertainty estimation (Figure 3) and results in suboptimal performance in downstream decision-making tasks such as multi-armed bandits and active learning (Theorem 3, Figure 3). We address this by using multi-step autoregressive generation in sequence models. Our findings are consistent with those of Zhang et al. [31], who also advocate for multi-step inference to improve active exploration via Thompson sampling.

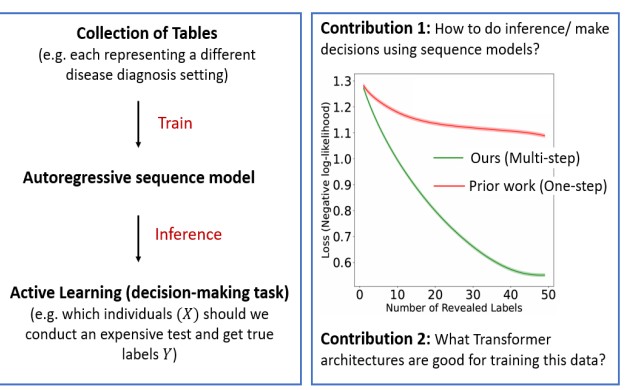

Figure 1: **(Left)** Meta-learned sequence models can be used for decision making for e.g. active learning **(Right)** Our contributions (1) multi-step outperforms one-step (2) Investigate which transformer architectures are good for training exchangeable data

This brings us to the next key question: What kind of architectures should we use for modeling exchangeable sequences, particularly when performing multi-step inference? Architectural choice is a crucial source of inductive bias, and existing approaches have attempted to incorporate exchangeability as an inductive bias through specialized masking strategies [22, 23, 30]. However, a critical gap in the literature remains: these prior works conflate exchangeability with the invariance properties enforced by their masking schemes that don't necessarily guarantee valid probabilistic inference.

**Another key contribution** of our work is in articulating this gap. We introduce conditional permutation invariance (Property 1) as a way to formalize the invariance enforced by these masking

strategies. Correcting previous works [22, 23, 30] that implicitly assume their architectures achieve exchangeability, we show that enforcing conditional permutation invariance alone is insufficient to guarantee full exchangeability. Specifically, existing masking-based approaches do not ensure the conditionally identically distributed (c.i.d.) property (Property 2), which is essential for the validity of probabilistic inference in exchangeable sequence models. As a result, despite their intended design, such models may still violate exchangeability when used in practice (see Section 4). By clearly distinguishing between exchangeability and the weaker notion of conditional permutation invariance, our work not only clarifies existing misconceptions in the literature but also establishes more rigorous foundation for exchangeable sequence models.

Moreover, a significant drawback of this masking scheme, as discussed in Section 4, is that it introduces computational overhead without yielding any tangible improvements in model performance. To evaluate its impact, we empirically assess the effectiveness of enforcing conditional-permutation invariance (Property 1) within the model architecture, comparing it to standard causal masking, which does not enforce permutation invariance. Surprisingly, our results show that enforcing Property 1 not only fails to provide any performance benefits but actually performs worse than causal masking (Figure 5). These findings underscore the need for new research directions to develop more effective inductive biases for Transformers in exchangeable sequence modeling.

## 2 Conceptual background

In this section, we first review *De Finetti's predictive view of Bayesian inference* [6, 7, 8, 9] and how autoregressive sequence modeling of exchangeable sequences can power it. For most of our discussion, we focus on the setting where observations are given by $Y_{1:\infty}$. However, this framework can be easily extended to contextual settings where observations are $(X_{1:\infty}, Y_{1:\infty})$, with $X$ serving as the context (Section C). We start by reviewing the conventional Bayesian modeling paradigm, where the modeler posits a latent parameter $\theta$, along with a prior $\mu(\theta)$, and likelihood $\mathbb{P}(Y_{1:\infty} \mid \theta) \equiv P_\theta(Y_{1:\infty})$. The joint probability of observations $Y_{1:\infty}$ is then expressed as $\mathbb{P}(Y_{1:\infty} = y_{1:\infty}) = \int \prod_{t=1}^\infty P_\theta(Y_t = y_t)\mu(d\theta)$.

Given any observable data $Y_{1:t}$ the posterior over latent parameter is expressed as $\mu(\cdot|Y_{1:t})$. Note that $\mu(\cdot|Y_{1:t})$ represents the epistemic (reducible) uncertainty, which gets resolved as more data is collected while the likelihood $\mathbb{P}(Y|\theta)$ represents the aleatoric uncertainty and comes due to inherent randomness in the data.

**Predictive view of uncertainty.** Instead of positing explicit priors and likelihoods over a proposed latent parameter space, we consider a different probabilistic modeling approach where we directly model the observable $Y_{1:\infty}$ without explicitly relying on any latent parameter. This view heavily relies on the **infinite exchangeability** of the sequence $Y_{1:\infty}$, defined as: $\mathbb{P}(Y_1, \cdots, Y_n) = \mathbb{P}(Y_{\pi(1)}, \cdots, Y_{\pi(n)})$, for any $n$ and permutation $\pi$. De Finetti's theorem states that if an infinite sequence is exchangeable then the sequence can be represented as a mixture of i.i.d. random variables.

**Theorem 1** (De Finetti's theorem). *If a sequence $Y_{1:\infty}$ is infinitely exchangeable then there exists a latent parameter $\theta$ and a unique measure $\mu(\cdot)$ over $\theta$, such that, for any n*

$$\mathbb{P}(Y_{1:n} = y_{1:n}) = \int \prod_{t=1}^n \mathbb{P}(Y_t = y_t|\theta)\mu(d\theta). \tag{1}$$

In addition to justifying conventional Bayesian modeling, De Finetti's theorem also establishes that for infinitely exchangeable sequence $Y_{1:\infty}$, the epistemic uncertainty in the latent parameter $\theta$ in Theorem 1 arises solely from the unobserved $Y_{1:\infty}$ [3, 14, 13, 16]. In other words, epistemic uncertainty in $\theta$ is the same as predictive uncertainty in $Y_{1:\infty}$. De Finetti [8], Hewitt and Savage [19] in fact show that the latent parameter $\theta$ in Equation (1) is entirely a function of the $Y_{1:\infty}$, that is, $\theta = f(Y_{1:\infty})$. To illustrate, consider coin $B$ from Figure 2 and suppose it is tossed repeatedly. Let $Y_t^{(B)}$ denote the outcome of $t$-th toss of coin $B$ where $Y_t^{(B)} = 1$ for heads, and $Y_t^{(B)} = 0$ for tails. The sequence $\{Y_1^{(B)}, \cdots, Y_\infty^{(B)}\}$ is exchangeable, since its probability distribution depends only on the outcomes of the tosses, not their *order*. In this case, De Finetti's Theorem holds trivially, and the latent parameter $\theta$ represents the probability of obtaining 'heads' when the coin $B$ is flipped. Further, this parameter $\theta$ can be estimated as $\theta = \lim_{t\to\infty} \frac{1}{t} \sum_{i=1}^t Y_i^{(B)}$.

By abusing notation and writing $\mathbb{P}(\cdot|Y_{1:\infty}) \equiv \mathbb{P}(\cdot|\theta)$ where $\theta = f(Y_{1:\infty})$, we can interpret $\mathbb{P}(\cdot|Y_{1:\infty})$ itself as the latent parameter $\theta$. Therefore, given some observation $Y_{1:t}$, generating $Y_{t+1:\infty} \sim \mathbb{P}(\cdot|Y_{1:t})$ is equivalent to sampling $\theta \sim \mu(\cdot|Y_{1:t})$. This indicates we can do equivalent Bayesian inference [12] using $\mathbb{P}(\cdot|Y_{1:t})$.

**Differentiating epistemic and aleatoric uncertainty.** Since epistemic uncertainty is equivalent to the predictive uncertainty of future observations, it can be reduced with additional observations. In contrast, *aleatoric* uncertainty refers to uncertainty that remains irreducible, even with more observations. For example, in the coin toss scenario (Figure 2), the uncertainty in Coin $A$ is aleatoric, arising from inherent randomness of a fair coin toss. No matter how many times coin $A$ is flipped, there will always be uncertainty about the outcome of the next coin $A$ toss. Coin $B$, on the other hand, has epistemic uncertainty which can be eliminated by flipping the coin once.

To quantify epistemic uncertainty and distinguish from aleatoric uncertainty, we must autoregressively generate $Y_{1:\infty} \sim \mathbb{P}(\cdot)$. As shown in Figure 2, considering the entire sequence of future coin tosses is crucial—examining only a single toss results in identical predictive uncertainty for both coins, making them indistinguishable. Formally, since $\mathbb{P}(Y|Y_{1:t}) = \int \mathbb{P}(Y|\theta)\mu(\theta|Y_{1:t})$, one-step prediction/generation fails to distinguish between epistemic and aleatoric uncertainty. We explore this further in Section 3.

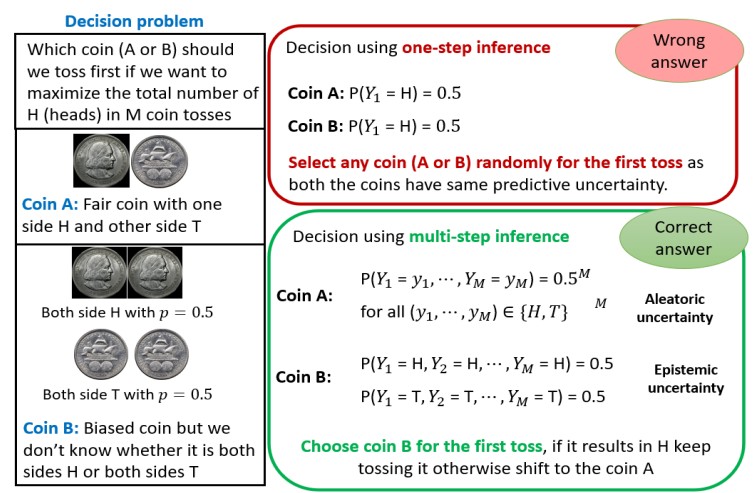

**Learning a probabilistic model from data.** We now shift our focus to directly learning $\mathbb{P}(Y_{1:T} = y_{1:T})$. We represent the transformer-based autoregressive sequence model, parametrized by $\phi$, as $\widehat{P}_\phi$. Rewrite $\widehat{P}_\phi(\hat{Y}_{1:T} = y_{1:T})$

Figure 2: **[Impact of Multi-step inference v/s One-step inference in decision making]** Coins A and B are considered identical by single-step inference because both have the same level of predictive uncertainty in their rewards. However, multi-step inference highlights a key difference: for Coin B, the uncertainty can be reduced (epistemic) by performing a single toss, whereas for Coin A, all the uncertainty is irreducible (aleatoric) and arises from the inherent randomness of a fair coin toss. Consequently, multi-step inference prioritizes tossing Coin B first to reduce epistemic uncertainty. (Illustration is adapted from [24]).

as an autoregressive product: $\prod_{i=0}^{T-1} \widehat{P}_\phi(\hat{Y}_{i+1} = y_{i+1} \mid \hat{Y}_{1:i} = y_{1:i})$, where $\widehat{P}(\hat{Y}_{i+1} \mid \hat{Y}_{1:i})$ represents a one-step predictive distribution, often referred to as posterior predictive in conventional Bayesian terminology. The transformer is trained to minimize the KL-divergence between the true data generating process $\mathbb{P}$ and the model $\widehat{P}_\phi$

$$D_{\mathrm{kl}}\left(\mathbb{P}\|\widehat{P}_\phi\right) \propto -\sum_{i=0}^{T-1} \mathbb{E}_{y_{i+1}\sim\mathbb{P}(\cdot|y_{1:i})}\left[\log\left(\widehat{P}_\phi\left(\hat{Y}_{i+1} = y_{i+1}|\hat{Y}_{1:i} = y_{1:i}\right)\right)\right].$$

Given a dataset of sequences $\{y_{1:T}^j : 1 \le j \le N\}$ generated from the true data generating process $\mathbb{P}(\cdot)$, we train the autoregressive sequence model (transformer) on this data using the following objective: $-\frac{1}{N}\sum_{j=1}^{N}\sum_{i=0}^{T-1}\log\widehat{P}_\phi(\hat{Y}_{i+1}^j = y_{i+1}^j \mid \hat{Y}_{1:i}^j = y_{1:i}^j)$. For simplicity, denote $\widehat{P}_\phi(\hat{Y}_{i+1} = y|\hat{Y}_{1:i})$ as $\widehat{P}_\phi^{(i+1)}(y)$. This training procedure enables the model to learn the single-step predictive distributions that collectively define the full sequence likelihood. Once trained, given any observed data $y_{1:t}$, the transformer sequence model can be used to generate future samples using one-step inference or multi-step inference. We now formally define one-step inference and multi-step inference. Suppose we want to predict $\hat{Y}_{t+1:T}$ given $y_{1:t}$:

**Definition 1** (One-step inference) Given $y_{1:t}$, we generate each $\widehat{Y}_i \sim \widehat{P}_\phi(\cdot|y_{1:t})$ i.i.d from the sequence model. That is, the probability of $\widehat{Y}_{t+1:T} = y_{t+1:T}$ is equal to $\prod_{i=t+1}^{T} \widehat{P}_\phi(\widehat{Y} = y_i|y_{1:t})$. We denote this distribution by $\widehat{P}_\phi^O(y_{t+1:T})$.

**Definition 2** (Multi-step inference) Given $y_{1:t}$, we generate $\widehat{Y}_{t+1:T}$ autoregressively from the sequence model. That is the probability of $\widehat{Y}_{t+1:T} = y_{t+1:T}$ is equal to $\prod_{i=t+1}^{T} \widehat{P}_\phi(\widehat{Y} = y_i|y_{1:i-1})$. We denote this distribution by $\widehat{P}_\phi^M(y_{t+1:T})$.

## 3 Inferential inductive biases for decision making

Much of the literature on autoregressive sequence modeling focuses on one-step inference or prediction. For example, recent work by Hegselmann et al. [18] empirically demonstrates the scalability of sequence models for training on tabular data, but their study emphasizes one-step prediction. Similarly, while Müller et al. [22], Nguyen and Grover [23], Hollmann et al. [20] discuss predictive uncertainty, their focus remains exclusively on one-step predictive uncertainty. As mentioned earlier in Section 2 and elaborated on later, one-step inference does not adequately differentiate between epistemic and aleatoric uncertainty. In contrast, multi-step inference provide a more effective means of quantifying epistemic uncertainty, which is crucial for sequential decision making applications [28, 24]. This distinction was illustrated informally in the coin toss example in Figure 2. To clarify this further, we present a more formal example in Section D.1.

### 3.1 Theoretical characterization of impact on decision-making

We emphasize the significance of the gap between single-step and multi-step inference by examining its impact on *decision-making*. Recall Figure 2 where we needed to decide whether to flip coin A or B first to maximize the total number of heads over $M$ coin tosses. One-step inference fails to distinguish between the two coins, leading to a suboptimal decision of selecting a coin randomly. In contrast, multi-step inference addresses this issue, enabling optimal decision-making.

**Characterizing information loss in one-step inference v/s multi-step inference:** We analyze how our uncertainty quantification—and inference in general—deteriorates when relying solely on one-step inference instead of multi-step inference. Let $y_{t+1:T} \sim \mathbb{P}(\cdot|y_{1:t})$ be some data from the data generating process. Further, let $\widehat{P}_\phi^O(y_{t+1:T}) \equiv \prod_{i=t+1}^{T} \widehat{P}_\phi(\widehat{Y} = y_i|y_{1:t})$ be the one-step inference model and let $\widehat{P}_\phi^M(y_{t+1:T}) \equiv \prod_{i=t+1}^{T} \widehat{P}_\phi(\widehat{Y} = y_i|y_{1:i-1})$ be the multi-step inference model, where we generate $\widehat{Y}_i$ autoregressively. Then the expected difference between the log-likelihood of the multi-step inference model and single-step inference model is given by the following result.

**Theorem 2.** *Assuming $\widehat{P}_\phi = \mathbb{P}$, then the difference $\mathbb{E}(\log[\widehat{P}_\phi^M(y_{t+1:T})] - \log[\widehat{P}_\phi^O(y_{t+1:T})])$ is equal to $\sum_{i=t+1}^{T} I(y_i; y_{t+1:i-1}|y_{1:t})$, where $I(A; B|C)$ is the mutual information between $A$ and $B$ conditional on $C$ and expectation is w.r.t. $y_{1:T} \sim \mathbb{P}(\cdot)$.*

This demonstrates that relying solely on one-step inference results in the loss of mutual information among $y_{t+1:T}$. Consequently, the expected likelihood under the multi-step inference model is higher than that under the one-step inference model and single-step inference is inherently less effective than multi-step inference. We further analyze expression $\sum_{i=t+1}^{T} I(y_i; y_{t+1:i-1}|y_{1:t})$ in a specific setting (Example 1) and show that as $\sigma^2$ (epistemic uncertainty) increases, the performance gap between one-step and multi-step inference widens, with one-step inference becoming increasingly suboptimal.

**Example 1.** Assuming $Y = \theta + \epsilon$ where $\epsilon \sim N(0, \tau^2)$ and $\theta \sim N(\mu, \sigma^2)$. Let $y_{t+1:T} \sim \mathbb{P}(\cdot|y_{1:t})$ and $(\sigma')^2 = \left(\frac{1}{\sigma^2} + \frac{t}{\tau^2}\right)^{-1}$, then expression $(\sum_{i=t+1}^{T} I(y_i; y_{t+1:i-1}|y_{1:t}))$ is equal to $\frac{1}{2} \log((1 + \frac{\sigma'^2}{\tau^2})^{T-t}) - \frac{1}{2} \log(1 + (T-t)\frac{\sigma'^2}{\tau^2})$.

**Characterizing impact on decision making:** We now examine how one-step inference affects downstream decision-making tasks. Specifically, we consider a one-armed bandit problem where the reward of the first arm is given by $Y^{(1)} \sim N(\theta, \tau^2)$ with $\theta \sim N(\mu, \sigma^2)$. While the second arm has a constant reward $Y^{(2)} = 0$. It is well known that Thompson sampling suffers $O(\sqrt{T})$ Bayesian regret in multi armed bandits [25]. We now consider the performance of Thompson sampling implemented

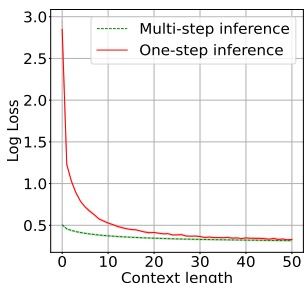
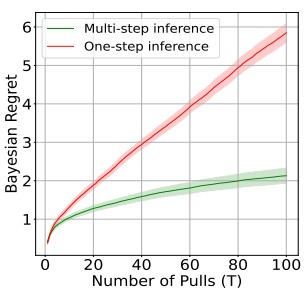
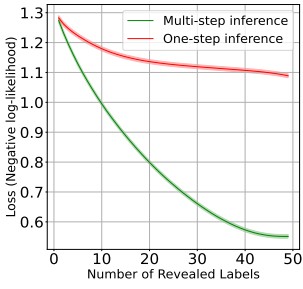

| (a) Uncertainty Quantification | (b) Multi-armed Bandits | (c) Active Learning |

Figure 3: **Comparing one-step inference and multi-step inference (lower is better): (a)** Uncertainty Quantification: Comparing multi-step log-loss for one-step and multi-step inference (Train horizon: $100$, Target Length: $50$) **(b)** Multi-armed Bandits: Comparing Bayesian regret under Thompson sampling algorithm using one-step and multi-step inference. **(c)** Active Learning: Comparing log-loss under uncertainty sampling algorithm using one-step and multi-step inference.

based on just the one-step inference or one-step inference (predictive uncertainty), with proof in Section D.

**Theorem 3.** *There exists one-armed bandit scenarios in which Thompson sampling incurs $O(T)$ Bayesian regret if it relies solely on one-step predictions from autoregressive sequence models.*

**Generalizing to the contextual setting:** We extend our analysis to the contextual setting in Section D, where the context $X \sim_{\text{i.i.d.}} P_X$. Additionally, we characterize the information loss associated with one-step inference in Bayesian linear regression and Gaussian processes in the same section.

## 3.2 Empirical investigation

In this section, we empirically evaluate the impact of single-step and multi-step inference on uncertainty quantification, as well as on downstream optimization tasks such as multi-armed bandits and active learning. We find that multi-step inference significantly outperforms one-step inference, being up to $60\%$ more efficient in bandit settings and requiring up to $10$ times less data in active learning to achieve the same predictive performance. [2]

### 3.2.1 Uncertainty quantification (UQ)

We evaluate one-step and multi-step inference by generating datasets using Gaussian Processes (GP), a common choice in prior works [22, 23]. Specifically, we employ a GP with an RBF kernel: $f \sim \mathcal{GP}(m, \mathcal{K})$, where $\mathcal{K}(X, X') = \sigma_f^2 \exp\left(-\|X - X'\|_2^2/2\ell^2\right)$. Additionally, Gaussian noise $N(0, \sigma^2)$ is added to the outputs. The input $X$ is drawn i.i.d. from $P_X$. To compare the performance of the two inference strategies, we use the multi-step log-loss metric. Further details on the metrics and experimental setup can be found in Section B. Figure 3(a) illustrates the comparison of multi-step log-loss performance between one-step and multi-step inference. Consistent with our theoretical results (Theorems 2 and 4), the results demonstrate that one-step inference performs worse than multi-step inference.

### 3.2.2 Multi-armed Bandits

**Problem:** We consider a two-armed Bayesian bandit setting with arms $\{C, D\}$ and $T$ rounds during which the arms are pulled. In each round $t$, based on the information collected so far, $\{(A_i, Y_i^{A_i}) : 1 \leq i \leq t - 1\}$, an arm $A_t$ is selected, and a reward $Y_t^{A_t}$ is observed. For each arm $a \in \{C, D\}$, the rewards are distributed as $Y_{1:T}^{(a)} \overset{\text{iid}}{\sim} N(\theta^{(a)}, (\tau^{(a)})^2)$, where the mean reward $\theta^{(a)}$ follows a prior distribution $\theta^{(a)} \sim N(\mu^{(a)}, (\sigma^{(a)})^2)$. The objective is to determine which arm $A_t \in \{C, D\}$ to pull in each round $t$ in order to minimize the Bayesian regret, defined as:

---

[2]Our code repository is available at: `https://github.com/namkoong-lab/Inductive-biases-exchangeable-sequence`.

$\mathbb{E}(\max_{a \in \{C,D\}}\{T\theta^{(a)}\} - \sum_{i=1}^{T} Y_t^{(A_t)})$, where expectation is taken over the randomness in the rewards ($Y$), the actions/policy ($A_t$), and the means ($\theta$).

**Training and Evaluation:** We train two transformers, one for each arm (C and D). To compare the two inference strategies, we first sample $\theta^* \sim \mu$ for each arm. We then implement Thompson Sampling (using the respective inference strategy) with the trained transformers to acquire rewards over a horizon $T$. The regret is evaluated as $[\max_{a \in \{C,D\}}\{T\theta^{*(a)}\} - \sum_{i=1}^{T} Y_t^{(A_t)}]$. Finally, we average the regret over 1000 different samples of $\theta^*$, with each run consisting of $T = 100$ steps, to compute the Bayesian regret. Additional details about training and algorithm are in Section B.

**Results:** Our results are summarized in Figure 3(b). As expected, cumulative regret increases with the number of pulls for both inference strategies. The figure also demonstrates that multi-step inference significantly outperforms one-step inference having upto $60\%$ less regret.

### 3.2.3 Active Learning

**Problem:** In active learning, the goal is to adaptively collect labels $Y$ for inputs $X$ to maximize the performance of a model $\psi(\cdot)$. We focus on a pool-based setting, where a pool of data points $\mathcal{X}^{pool}$ is given, and the objective is to sequentially query labels $Y$ for $X \in \mathcal{X}^{pool}$. We consider a regression setting in which inputs $X \overset{\text{iid}}{\sim} P_X$, and outcomes are generated from an unknown function $f^*$, such that $Y = f^*(X) + \epsilon_X$, where the noise $\epsilon_X \sim N(0, \tau_X^2)$ is heteroscedastic. Additionally, the data-generating function $f^*$ drawn from a distribution $\mu$.

**Training and Evaluation:** We consider a meta learning setup where we train the sequence model (transformer) on data $\{(X_{1:N}^{(j)}, Y_{1:N}^{(j)}) : j \in [1, M]\}$ generated from the original data generating process. To evaluate the two inference strategies, we first sample $f^* \sim \mu$ and generate a dataset $\mathcal{X} \times \mathcal{Y} \equiv \mathcal{D}$, which contains both pool dataset ($\mathcal{D}^{pool}$) and test dataset ($\mathcal{D}^{test}$). Using the trained transformer and the respective inference strategy for uncertainty sampling, we sequentially select $X \in \mathcal{X}^{pool}$ for which labels are queried. At each time step $t$, given the collected data $\mathcal{D}^t \subset \mathcal{D}^{pool}$, we evaluate the transformer model's performance as $[-\sum_{(X,Y) \in \mathcal{D}^{test}} \widehat{P}_\phi(Y|X, \mathcal{D}^t)]$.

**Results:** The results are summarized in Figure 3(c). For both inference strategies, prediction accuracy improves, and loss decreases as more data points are acquired. However, multi-step inference significantly outperforms single-step inference, achieving the same performance level with nearly 10 times fewer samples.

Now that we have established the importance of multi-step inference, the next key question arises: What architectures are best suited for modeling exchangeable sequences, especially when performing multi-step inference?

## 4 Architectural inductive biases

Ensuring that the sequence model $\widehat{P}_\phi$ is infinitely exchangeable enables robust performance and reliable statistical inference [30]. Several prior works have proposed architectural approaches to enforce exchangeability. Müller et al. [22] introduced a masking scheme designed to enforce exchangeability (Figure 4). In this scheme, all context points attend to one another, allowing the model to condition on the entire context set without any predefined ordering constraints. This design ensures that the model's predictions remain invariant to the order of context points, aligning with the principle that exchangeable sequences should not depend on the specific order in which observations are presented. Figure 4 illustrates the masking scheme corresponding to this architecture, where $(x_1, y_1), \cdots, (x_3, y_3)$ denote the context points, and $(x_4, 0)$ represents the target point for which $\hat{y}_4$ is to be predicted. Nguyen and Grover [23] proposed a similar architecture with modifications to improve training effi-

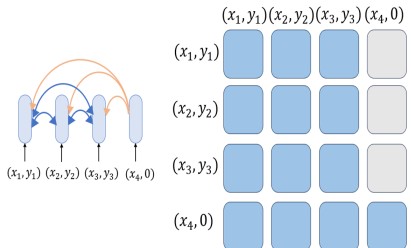

Figure 4: A representative attention mechanism and masking scheme widely used in prior literature to enforce exchangeability. However, it only ensures the **conditionally permutation-invariant** property.

ciency; however, their design contained an error where the target point did not attend to itself. This issue was later corrected by Ye and Namkoong [30].

Despite these efforts, the exchangeability guarantees of these architectures remain uncertain. Previous works implicitly assume that their masking schemes enforce exchangeability but fail to provide a formal characterization of what is actually achieved. To address this gap, we introduce a precise definition that explicitly captures the invariance imposed by these architectures. Our analysis reveals that all transformer-based architectures in the literature [22, 23, 30] aim to enforce a *"conditional permutation invariance property,"* which we formally define below.

**Property 1.** $\widehat{P}_\phi$ *is conditionally permutation invariant if* $\widehat{P}_\phi(\hat{Y}_{t+1}|\hat{Y}_{\pi(1)}, \cdots, \hat{Y}_{\pi(t)}) = \widehat{P}_\phi(\hat{Y}_{t+1}|\hat{Y}_1, \cdots, \hat{Y}_t)$.

This property ensures that the predictive uncertainty in $y_{t+1}$ remains the same under any permutation of the context $y_{1:t}$. We refer the reader to Section C for the corresponding definitions in contextual settings. Additionally, in Section 4.1, we provide a detailed discussion of the transformer architecture with the conditional permutation invariance property, including its efficient training procedure and the computational requirements for inference.

However, while conditional permutation invariance is a necessary characteristic of exchangeability, it is not sufficient. Enforcing this property alone does not guarantee full exchangeability—a critical oversight in prior work. For example, all exchangeable sequence models must also satisfy another crucial property called the conditionally identically distributed (c.i.d.) property, also known as the martingale property:

**Property 2.** *Recalling* $\widehat{P}_\phi^t(y) \triangleq \widehat{P}_\phi(\hat{Y}_t = y \mid \hat{Y}_{1:t-1})$, $\widehat{P}_\phi$ *is conditionally Identically Distributed (c.i.d.) if* $\mathbb{E}[\widehat{P}_\phi^{t+1}(y) \mid \hat{Y}_{1:t-1}] = \widehat{P}_\phi^t(y)$.

This property ensures that the expected predictive distribution at time $t + 1$, given past observations ($\hat{Y}_{1:t-1}$), is consistent with the predictive distribution at time $t$. The importance of c.i.d. property in exchangeable sequence models is what powers their ability to quantify epistemic uncertainty, as emphasized by previous work in Bayesian statistics [2, 4, 12, 11]. However, an autoregressive sequence model that satisfies Property 1 does not necessarily satisfy Property 2. We demonstrate this with a concrete example provided in Appendix D.2.

Although we have established that C-permutation invariant architectures do not achieve full exchangeability, it is still important to assess whether incorporating Property 1 into transformer architectures offers any advantages. To investigate this, we compare it to the standard causal transformer architecture (Section 4.2), which does not exhibit this property. Specifically, we analyze two architectures: (1) the conditionally permutation-invariant architecture and (2) the standard causal masking architecture. We describe their respective masking schemes, efficient training procedures, and computational requirements for inference.

### 4.1  Conditionally permutation invariant architecture

As we have established, achieving conditional permutation invariance requires ensuring that all context points can attend to one another. We adopt the same masking scheme shown in Figure 4, which, as previously discussed, has also been utilized in prior works such as [22, 20].

**An efficient training procedure:** Training directly in a naive manner can be inefficient, as it would require processing $T$ separate sequences, each of length $i \in \{1, 2, \cdots, T\}$, for an sequence data of length $T$. A more efficient approach is to fix a context length $i$ and train the transformer on multiple target points simultaneously. This process is then repeated for all possible context lengths $i$. The corresponding masking scheme is shown in Figure 7. Importantly, this method is equivalent to training for $P(y|x_{1:4}, y_{1:4}, x)$ across multiple values of $x$ in parallel, solely to optimize training efficiency. However, during inference, a multi-step (autoregressive) prediction approach should be adopted, as described in Section 3.

**Inference compute:** Predicting a sequence of length $T$ requires approximately $O(T^3)$ computational effort. A major drawback of this masking scheme is that, even with KV caching, the compute cannot be reduced to $O(T^2)$. This limitation arises because, at each inference step, all outputs from the attention heads must be recomputed as every point attends to the newly added context point (see

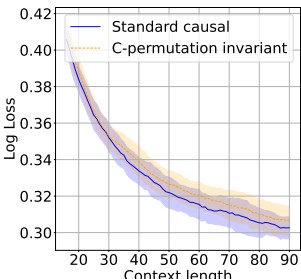
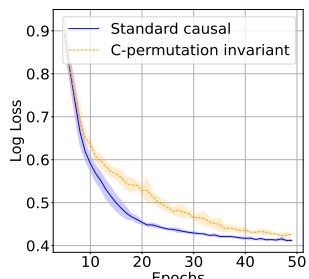
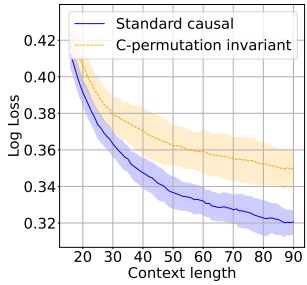

**(a)** In-training horizon performance of two architectures

**(b)** Training/data efficiency of two architectures

**(c)** Out-of-training horizon performance of two architectures

Figure 5: **Comparing C-permutation invariant architecture and Standard causal architecture (lower is better): (a)** In-training horizon performance **(b)** Training/Data efficiency **(c)** Out-of-training horizon performance

Figure 9 for details). This behavior contrasts with the typical causal masking approach, where such recomputation is avoided.

## 4.2 Standard causal architecture

In this scheme, each context point attends only to the previous context points and not to any future points. Specifically, the $i^{th}$ context point attends only to the context points within $[1 : i - 1]$. In the interest of space, the masking scheme for this architecture is illustrated in Figure 6 (Section A).

**An efficient training procedure:** As before, an efficient training procedure involves fixing a context length $i$ and training the transformer on multiple target points simultaneously. The corresponding masking scheme is shown in Figure 8.

**Inference compute:** Predicting a sequence of length $T$ requires approx. $O(T^3)$ computational effort. However, with KV caching, this can be reduced to $O(T^2)$ (see Figure 10).

## 4.3 Empirical investigation

In this section, we compare the contextual permutation invariance architecture and standard causal architectures. As is common in the literature [22, 23, 24], we use the standard log-loss metric to compare these architectures and defer the analysis of downstream performance to Section E. Our findings indicate that the in-training horizon performance of the C-permutation-invariant architecture is comparable to that of the standard causal architecture. However, the standard causal architecture outperforms the C-permutation-invariant model on the out-of-training horizon by approximately $10\%$ and demonstrates greater training and data efficiency by up to $20\%$. Furthermore, as noted earlier in Section 4, the standard causal architecture requires significantly less inference computation due to benefits from KV caching. To evaluate these architectures we use same data generating process as in Section 3.2.1.

**Evaluation Metric:** To compare these two architectures, we use two metrics - one-step log-loss and multi-step log-loss. These metrics are described in detail in Section B.

Both masking schemes are trained on data generated from Gaussian Processes. Additional details about the architectures and training process are provided in Section B. For brevity, we present only the results based on multi-step log-loss in the main body, while results for one-step log-loss are deferred to Section E.

**Results:** We evaluate these architectures across three dimensions - performance on sequence lengths within the training horizon, and beyond the training horizon, training/data efficiency.

*1. In-training horizon performance:* Figure 5(a) shows that the performance of both masking schemes is comparable, with no evident advantage of enforcing conditional-permutation invariance. Furthermore, additional ablation studies (see Section E) indicate that standard-causal masking may outperform conditional-permutation invariance masking.

*2. Data/Training efficiency:* To evaluate the data/training efficiency of these architectures, we compare their performance across various training epochs. Figure 5(b) indicates that standard causal masking exhibits superior training efficiency. Similar findings were consistently observed in the ablation studies (Section E).

*3. Out-of-training horizon performance:* To assess this performance, we train both architectures up to a horizon $m = 15$ and evaluate their performance on horizons beyond $m = 15$. Figure 5(c), along with additional experiments (see Section E), suggests that the standard causal masking may achieve better performance on out-of-training horizons.

## 5 Conclusion and future work

We empirically and theoretically demonstrate that one-step inference using sequence models, which has been the primary focus of the literature thus far, is insufficient for distinguishing between epistemic and aleatoric uncertainty, ultimately leading to suboptimal decision-making. In contrast, multi-step autoregressive generation effectively overcomes this limitation, as evidenced by empirical results in multi-armed bandit and active learning settings. On the architectural side, much of the existing work has focused on enforcing a masking scheme that, as we identify, only ensures conditional permutation invariance in transformers rather than full exchangeability. Empirically, we find that this approach performs worse than standard causal masking and significantly increases the computational cost of multi-step inference, as it cannot leverage KV caching. A limitation of our work is that we were unable to identify a transformer architecture that achieves both full exchangeability and computational efficiency for decision-making and active exploration. Developing such an architecture remains an important direction for future research.

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

# A   Additional Demonstrations

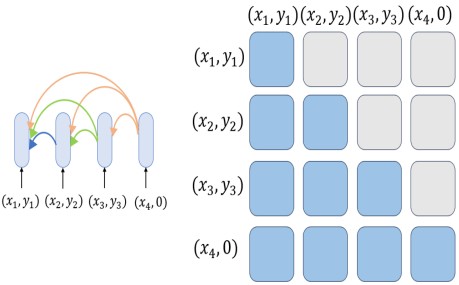

Figure 6: **Standard causal** transformer architecture: attention mechanism and masking scheme

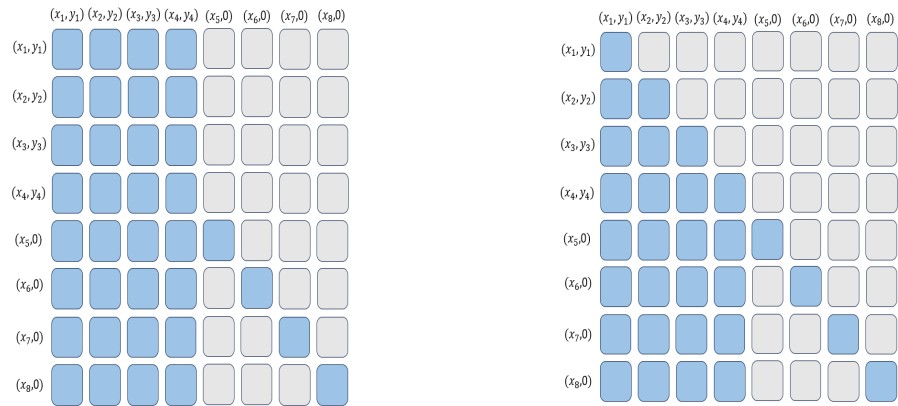

Figure 7: Efficient training procedure for conditionally permutation invariant transformer

Figure 8: Efficient training procedure for standard causal transformer architecture

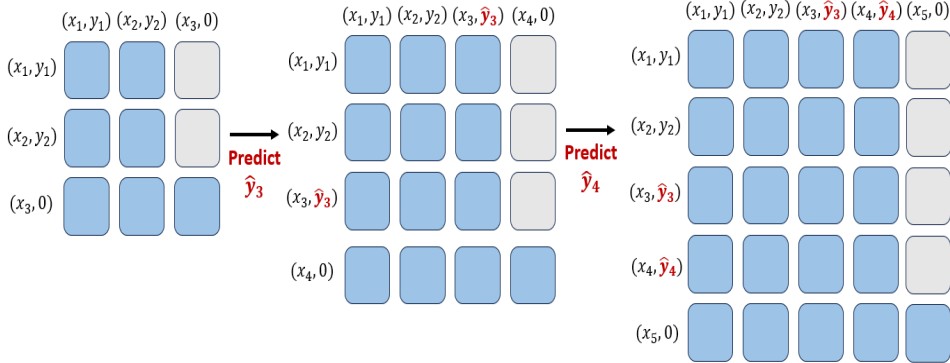

Figure 9: Conditional permutation invariant architecture - masking scheme at inference stage

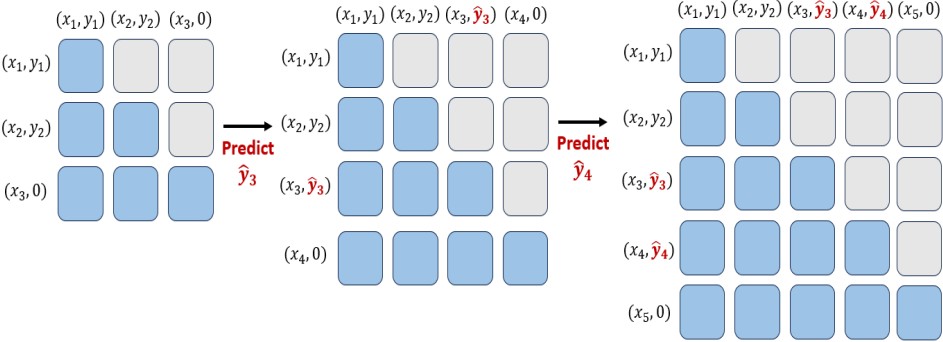

Figure 10: Standard causal architecture - masking scheme at inference stage

# B  Experiments Details

In this section, we present the experimental details from Sections 3.2 and 4.3.

## B.1  Details for experiments in Section 3.2.1 and 4.3

**Data generating process:**    As previously mentioned, we generate data synthetically using Gaussian processes. Specifically, we employ a Gaussian Process (GP) with a Radial Basis Function (RBF) kernel: $f \sim \mathcal{GP}(m, \mathcal{K})$, where $m(X)$ represents the mean function, and $\mathcal{K}(X, X') = \sigma_f^2 \exp\left(-\frac{||X-X'||_2^2}{2\ell^2}\right)$ represents the covariance function. Additionally, Gaussian noise $N(0, \sigma^2)$ is added to the outputs. The input $X$ is drawn i.i.d. from $P_X$. Unless stated otherwise, the parameters are set as follows: $m(X) = 0$, $X \sim U[-2.0, 2.0]$, $\sigma_f = 1.0$, $\ell = 1.0$, $\sigma = 0.1$.

**Evaluation Metric:**    To compare these two architectures, we employ two metrics: one-step log-loss and multi-step log-loss. A detailed description of these metrics is provided below.

1. *One-step log-loss:* Given $(x_{1:m}, y_{1:m}, x_{m+1})$, we generate $y_{m+1}$ from the true data generating process, i.e., $y_{m+1} \sim \mathbb{P}(\cdot|x_{m+1}, x_{1:m}, y_{1:m})$. The one-step log-loss for the sequence model is then calculated as: $-\log\left[\widehat{P}_\phi(y_{m+1}|x_i, x_{1:m}, y_{1:m})\right]$. This value is further averaged over multiple instances of $(x_{1:m}, y_{1:m}, x_{m+1})$.

2. *Multi-step log-loss:* Given $(x_{1:m}, y_{1:m}, x_{m+1:T})$, we generate $y_{m+1:T}$ from the true data-generating process, i.e., $y_{m+1:T} \sim \mathbb{P}(\cdot|x_{m+1:T}, x_{1:m}, y_{1:m})$. The multi-step log-loss for the sequence model is computed as: $-\log\left[\widehat{P}_\phi(y_{m+1:T}|x_{m+1:T}, x_{1:m}, y_{1:m})\right]$, which can be expressed as: $-\sum_{i=m}^{T-1} \log\left[\widehat{P}_\phi(y_{i+1}|x_{m+1:i+1}, x_{1:m}, y_{1:m}, y_{m:i})\right]$. This value is then averaged over multiple instances of $(x_{1:m}, y_{1:m}, x_{m+1:T})$.

We refer to $t$ as the *context-length* and $(T - t)$ as the *target length*.

**Transfromer architecture and training details:**    To compare the conditionally permutation-invariant architecture with the standard causal architecture, we use a decoder-only transformer with the following parameters. Both architectures share the same parameters, differing only in their masking schemes. The model parameters are as follows:

- Model dimension: 64
- Feedforward dimension: 256
- Number of attention heads: 4
- Number of transformer layers: 4
- Dropout: 0.1

- Activation function: GELU

For embedding $(x, y)$, we use a neural network with two layers of sizes $[256, 64]$. Additionally, a final linear layer is used to predict the mean $\mu$ and standard deviation $\sigma$ of the output distribution, modeled as $Y \sim N(\mu, \sigma^2)$. For training the transformers, we use the Adam optimizer with default parameters, and the learning rate is adjusted using a cosine scheduler. The training parameters are as follows: Warmup ratio is $0.03$, minimium learning rate is $3.0e^{-5}$, learning rate is $0.0003$, weight decay to $0.01$ and batch size is $64$. For all the experiments we train the transformer for 400 epochs.

**Computational resources:** We use NVIDIA A100-SXM4-80GB for training our models. For the standard-causal architecture it takes 4hr, while for C-permutation variant architecture it takes 17hr to train the model.

**Evaluation:** We evaluate the trained transformers, corresponding to each architecture, using one-step log-loss and multi-step log-loss. For each experiment, we use $8192$ test samples and conduct evaluations across five different random seeds. This process includes retraining the models on different training datasets and evaluating them on distinct test datasets.

## B.2 Experimental details for Section 3.2 - Bayesian Bandits

**Data generating process:** Recall that, for each arm $a \in \{C, D\}$, the rewards are distributed as $Y_{1:T}^{(a)} \overset{\text{iid}}{\sim} N\left(\theta^{(a)}, \left(\tau^{(a)}\right)^2\right)$, where the mean reward $\theta^{(a)}$ follows a prior distribution $\theta^{(a)} \sim N\left(\mu^{(a)}, \left(\sigma^{(a)}\right)^2\right)$. Where for Arm $C$, we set $\mu^{(C)} = 0$, $\sigma^{(C)} = 0.5$ and $\tau^{(C)} = 0.5$. While for Arm $D$, we set $\mu^{(D)} = 0$, $\sigma^{(D)} = 0.9$ and $\tau^{(D)} = 0.1$.

**Exploration Algorithm:** Numerous algorithms have been proposed in the literature to address this trade-off, such as Thompson Sampling [25], Bayes-UCB [21] and Gittin's index [27]. However, to compare the two inference strategies, we fix the algorithm to Thompson Sampling and implement it using the respective strategy.

**Transformer Architecture and Training:** As specified earlier for each arm $a \in \{C, D\}$, we train a separate transformer (decoder-only). The model parameters and training details are specified below:

- Model dimension: 64
- Feedforward dimension: 256
- Number of attention heads: 4
- Number of transformer layers: 4
- Dropout: 0.1
- Activation function: GELU

For embedding $(x, y)$, we use a neural network with two layers of sizes $[256, 64]$. Note that, as in this case there is no context, we set $x$ as 0. Additionally, a final linear layer is used to predict the mean $\mu$ and standard deviation $\sigma$ of the output distribution, modeled as $Y \sim N(\mu, \sigma^2)$. For training the transformers, we use the Adam optimizer with default parameters, and the learning rate is adjusted using a cosine scheduler. The training parameters are as follows: Warmup ratio is $0.03$, minimium learning rate is $3.0e^{-5}$, learning rate is $0.0003$, weight decay to $0.01$ and batch size is $64$. For all the experiments we train the transformer for 400 epochs.

**Evaluation Details:** The results presented in Figure 3(b) are averaged over $1000$ different experiments, with each experiment run for $100$ steps.

**One-step / Multi-step inference (Thompson sampling):** In Algorithm 1, we outline the implementation of a Thompson sampling algorithm for one-step and multi-step inference using a trained transformer. In Algorithm 1 setting $J = 1$ corresponds to one-step inference, while choosing $J > 1$ enables multi-step inference. In our experiments, we set $J = 100$ for multi-step inference. For reference, we also provide the standard Thompson sampling algorithm for Gaussian-Gaussian multi-armed bandits (see Algorithm 2). In our experiments, we set $J = 100$ for multi-step inference.

---

**Algorithm 1** One-step and Multi-step inference (Thompson sampling) using sequence models (transformers) in multi armed bandits setting

---

**Require:** Trained transformer $(\widehat{P}_\phi^{(a)})$ for each of the Arm $a \in [1, K]$, horizon $T$, Number of autoregressive generations in each iteration $(J)$ : In one-step inference $J = 1$, while for multi-step inference $J > 1$.

1: **Initialization:**
2: **for** $a = 1 \to K$ **do**
3:     $\mathcal{Y}^{(a)} \leftarrow \{\}$                                    // Observed rewards for each arm $a$
4:     $n^{(a)} \leftarrow 0$                                       // number of pulls for arm $a$
5: **end for**
6: **for** $t = 1 \to T$ **do**
7:     **Autoregressively** generate J rewards for each arm, conditioned on observed rewards
8:     **for** $j = 1 \to J$ **do**
9:

$$\hat{Y}_{t,n^{(a)}+j}^{(a)} \sim \widehat{P}_\phi(\cdot | \mathcal{Y}^{(a)}, \hat{Y}_{t,n^{(a)}+1}^{(a)}, \cdots, \hat{Y}_{t,n^{(a)}+j-1}^{(a)})$$

    // In $\hat{Y}_{t,n^{(a)}+1}^{(a)}$, $t$ indicates time-step and $n^{(a)} + j$ indicates conditioning on $|\mathcal{Y}^{(a)}| = n^{(a)}$
    observations and $j - 1$ generations.
10:     **end for**
11:     **Select** arm $A_t = \arg \max_{1 \leq a \leq K} \frac{1}{J} \sum_{j=1}^{J} \hat{Y}_{t,n^{(a)}+j}^{(a)}$.

12:     **Pull** arm $A_t$ and **observe** reward $Y_t^{(A_t)}$.
13:     **Update** the collection of observations for arm $A_t$:

$$n^{(A_t)} \leftarrow n^{(A_t)} + 1, \quad \mathcal{Y}^{(A_t)} \leftarrow \mathcal{Y}^{(A_t)} \cup \{Y_t^{(A_t)}\}.$$

14: **end for**

---

### B.3 Experimental details for Section 3.2 - Active Learning

**Data generating process:** Recall that our data generating process is as follows - features $X \overset{\text{iid}}{\sim} P_X$, and outcomes are generated from an unknown function $f^*$, such that $Y = f^*(X) + \epsilon_X$, with noise $\epsilon_X \sim N(0, \tau_X^2)$ being heteroscedastic and the data-generating function $f^*$ drawn from a distribution $\mu$. Our $P_X$ consists of 100 *non-overlapping clusters*. Further, within each cluster $f^*(X)$ is highly correlated, while across clusters, the correlation is low. Further on some clusters the noise $\epsilon_X$ is high, while on others it is low. This setting introduces a setting where it is necessary to differentiate between aleatoric and epistemic noise for efficiently querying the labels for model improvement.

**Exploration Algorithm:** There are various active learning query strategies, such as uncertainty sampling techniques (e.g., margin-sampling or entropy) and Bayesian Active Learning by Disagreement (BALD) [26, 1]. For this study, we focus on Uncertainty Sampling adapted to the regression setting.

**Transformer Architecture and Training:** We train a decoder-only transformer on sequential-data $\{(X_{1:N}^{(j)}, Y_{1:N}^{(j)}) : j \in [1, M]\}$ generated from the original data generating process. The model parameters and training details are specified below.

- Model dimension: 64
- Feedforward dimension: 256
- Number of attention heads: 4
- Number of transformer layers: 4
- Dropout: 0.1
- Activation function: GELU

For embedding $(x, y)$, we use a neural network with two layers of sizes $[256, 64]$. Additionally, a final linear layer is used to predict the mean $\mu$ and standard deviation $\sigma$ of the output distribution,

---

**Algorithm 2** Thompson Sampling for Multi Armed Bandits (Gaussian-Gaussian setting)

---

**Require:** Number of arms $K$, prior mean $\mu^{(a)}$, prior variance $\left(\sigma^{(a)}\right)^2$, known reward variance $\left(\tau^{(a)}\right)^2$, horizon $T$.

1: **Initialization:**
2: **for** $a = 1 \rightarrow K$ **do**
3:     $n^{(a)} \leftarrow 0$                                  // number of pulls for arm $a$
4:     $S^{(a)} \leftarrow 0$                                 // sum of rewards for arm $a$
5:     $\hat{\mu}^{(a)} \leftarrow \mu^{(a)}$                              // posterior mean for arm $a$
6:     $\left(\hat{\sigma}^{(a)}\right)^2 \leftarrow \left(\sigma^{(a)}\right)^2$             // posterior variance for arm $a$
7: **end for**
8: **for** $t = 1 \rightarrow T$ **do**
9:     **Sample** a mean from each arm's posterior:

$$\tilde{\mu}_t^{(a)} \sim \mathcal{N}\left(\hat{\mu}^{(a)}, \left(\hat{\sigma}^{(a)}\right)^2\right) \quad \text{for } a = 1, \ldots, K.$$

10:     **Select** arm $A_t = \arg\max\limits_{1 \leq a \leq K} \tilde{\mu}_t^{(a)}$.

11:     **Pull** arm $A_t$ and **observe** reward $Y_t^{(A_t)}$.
12:     **Update** the sufficient statistics for arm $A_t$:

$$n^{(A_t)} \leftarrow n^{(A_t)} + 1, \quad S^{(A_t)} \leftarrow S^{(A_t)} + Y_t^{(A_t)}.$$

13:     **Update** posterior for arm $A_t$.

$$\text{Posterior mean: } \hat{\mu}^{(A_t)} \leftarrow \frac{\frac{1}{\left(\sigma^{(A_t)}\right)^2}\mu^{(A_t)} + \frac{S^{(A_t)}}{\sigma^2}}{\frac{1}{\left(\sigma^{(A_t)}\right)^2} + \frac{n^{(A_t)}}{\tau^2}}$$

$$\text{Posterior variance: } \left(\hat{\sigma}^{(A_t)}\right)^2 \leftarrow \left(\frac{1}{\left(\sigma^{(A_t)}\right)^2} + \frac{n^{(A_t)}}{\tau^2}\right)^{-1}.$$

14: **end for**

---

modeled as $Y \sim N(\mu, \sigma^2)$. For training the transformers, we use the Adam optimizer with default parameters, and the learning rate is adjusted using a cosine scheduler. The training parameters are as follows: Warmup ratio is $0.03$, minimum learning rate is $3.0e^{-5}$, learning rate is $0.0003$, weight decay to $0.01$ and batch size is $64$. For all the experiments we train the transformer for $400$ epochs.

**Evaluation Details:** Results shown in Figure 3(c) are averaged over $50000$ experiments with each experiment run for $50$ steps.

**One-step / Multi-step inference (Uncertainty sampling):** In Algorithm 3 we describe the implementation of one-step and multi-step inference based uncertainty sampling using trained transformers. In our multi-step inference experiments, we set $= J = 20$ and $I = 20$.

## C  Contextual setting definitions

Consider a contextual setting, where the context $X \overset{\text{iid}}{\sim} P_X$.

**Exchangeability definition:** An infinite sequence $(X_{1:\infty}, Y_{1:\infty})$ is exchangeable if .for any $n$ and permutation $\pi$

$$\mathbb{P}((X_1, Y_1), \cdots, (X_n, Y_n)) = \mathbb{P}((X_{\pi(1)}, Y_{\pi(1)}), \cdots, (X_{\pi(n)}, Y_{\pi(n)})).$$

**Algorithm 3** One-step and Multi-step inference (Uncertainty sampling) using sequence models (transformers) in active learning setting

---

**Require:** Trained transformer $(\widehat{P}_\phi)$, Horizon $T$, Initial data available $\mathcal{D}^0$, Pool to choose from $\mathcal{X}^{pool}$. Number of autoregressive generations in each iteration $(J)$ : In one-step inference $J = 1$, while for multi-step inference $J > 1$; Number of generation paths over which variance is taken $I$.

1: **for** $t = 1 \to T$ **do**
2:    **for** $X \in \mathcal{X}^{pool}$ **do**
3:       Generate $I$ trajectories for each sample $X$ and collect the mean output on each trajectory in list $l_{t,X}$.
4:       Initialize $l_{t,X} = \{\}$
5:       **for** $i = 1 \to I$ **do**
6:          **Autoregressively** generate J sample outputs for $X$ for each arm, conditioned on the available data.
7:          **for** $j = 1 \to J$ **do**
8:

$$\hat{Y}_{t,j}^{(i,X)} \sim \widehat{P}_\phi(\cdot|\mathcal{D}^{t-1}, (X, \hat{Y}_{t,1}^{(i,X)}), \cdots, (X, \hat{Y}_{t,j-1}^{(i,X)}), X)$$

9:          **end for**
10:          $l_{(t,X)} \leftarrow l_{(t,X)} \cup \{\frac{1}{J}\sum_{j=1}^{J} \hat{Y}_{t,j}^{(i,X)}\}$
11:       **end for**
12:       Estimate variance of the mean output across $I$ trajectories $\hat{V}_{(t,X)} = Variance_{\bar{Y} \in l_{(t,X)}}(\bar{Y})$
13:    **end for**
14:    **Select** $X^* = \arg \max_{X \in \mathcal{X}^{pool}} \hat{V}_{t,X}$.
15:    **Query** $X^*$ and get the true label/output $Y^*$.
16:    **Update** $\mathcal{D}^t \leftarrow \mathcal{D}^{t-1} \cup (X^*, Y^*)$
17: **end for**

---

**Transformer training:**

$$\min_\phi \left\{ -\frac{1}{N} \sum_{j=1}^{N} \sum_{i=0}^{T-1} \log \widehat{P}_\phi \left( \hat{Y}_{i+1}^j = y_{i+1}^j \mid \hat{Y}_{1:i}^j = y_{1:i}^j, x_{1:i}^j, x_{i+1}^j \right) \right\}.$$

**Conditional permutation invariance property:**

$$\widehat{P}_\phi(Y_{t+1}|(X_1, Y_1), \cdots, (X_t, Y_t), X_{t+1}) = \widehat{P}_\phi(Y_{t+1}|(X_{\pi(1)}, Y_{\pi(1)}), \cdots, (X_{\pi(t)}, Y_{\pi(t)}), X_{t+1}).$$

**Conditionally identically distributed property:** In the presence of covariates, where $X \sim P_X$ independently, the c.i.d. property extends to the sequence model as follows:

$$\mathbb{E}\left( \widehat{P}_\phi^{t+1}(y \mid x) \mid \hat{Y}_{1:t-1}, X_{1:t-1} \right) = \widehat{P}_\phi^t(y \mid x).$$

where $\quad \widehat{P}_\phi\left( \hat{Y}_t = y \mid X_t = x, \hat{Y}_{1:t-1}, X_{1:t-1} \right) \quad =: \quad \widehat{P}_\phi^t(y \quad \mid \quad x) \quad$ and $\widehat{P}_\phi\left( \hat{Y}_{t+1} = y \mid X_{t+1} = x, \hat{Y}_{1:t}, X_{1:t} \right) =: \widehat{P}_\phi^{t+1}(y \mid x)$.

## D   Examples and Proofs

### D.1   Example differentiating epistemic and aleatoric uncertainty

**Example 2.** Consider a setting where the observation $Y \sim P_\theta = N(\theta, \tau^2)$ follows a normal distribution with mean $\theta \sim \mu = N(a, \sigma^2)$. Given observations $y_{1:t}$, the posterior distribution $\mu(\theta|y_{1:t}) = N(a_t, \sigma_t^2)$ represents the epistemic (reducible) uncertainty, while $Y \sim N(\theta, \tau^2)$ captures the aleatoric (irreducible) uncertainty caused by inherent randomness in the data.

*One-step inference* corresponds to $\mathbb{P}(Y|y_{1:t})$, which is equivalent to $\int P_\theta(Y)\mu(\theta|y_{1:t})d\theta = N(a_t, \sigma_t^2 + \tau^2)$. In this case, the predictive uncertainty for $Y$ (characterized by $\sqrt{\sigma_t^2 + \tau^2}$), combines both epistemic ($\sigma_t^2$) and aleatoric ($\tau^2$) uncertainties. Importantly, this approach does not allow us to separate the two components.

*Multi step inference,* on the other hand involves $\mathbb{P}(Y_{t+1:\infty}|Y_{1:t})$, which is (by De Finneti's) equivalent to $\int \prod_{i=t+1}^\infty P_\theta(Y_i)\mu(\theta|y_{1:t})d\theta$. Furthermore, as $T \to \infty$, we have $\lim_{T\to\infty} \frac{1}{T}\sum_{i=t+1}^T Y_i = \theta$ where $\theta \sim \mu(\cdot|y_{1:t}) = N(a_t, \sigma_t^2)$. Hence, the variance of the long-term average $\frac{1}{T}\sum_{i=t+1}^T Y_i$ is approximately $\sigma_t^2$ - isolates the epistemic uncertainty. Therefore, multi-step inference enables the differentiation between epistemic ($\sigma_t^2$) and aleatoric ($\tau^2$) uncertainties.

## D.2  Example demonstrating Property 1 does not lead to Property 2 and exchangeability

**Example 3.** Suppose $\widehat{P}_\phi^1(0) := \widehat{P}_\phi(0) = \frac{1}{3}$. Further, conditioned on first observation, assume that $\widehat{P}_\phi^1(1) := \widehat{P}_\phi(1) = \frac{2}{3}$ $\widehat{P}_\phi^2(0) := \widehat{P}_\phi(0|\hat{Y}_1) = \frac{1}{3}$; $\widehat{P}_\phi^2(1) := \widehat{P}_\phi(1|\hat{Y}_1) = \frac{2}{3}$. Observe that $Y_1$ and $Y_2$ are i.i.d. Finally, conditioned on first two observations, assume that $\widehat{P}_\phi^3(0) := \widehat{P}_\phi(0|\hat{Y}_1, \hat{Y}_2) = \frac{\hat{Y}_1 + \hat{Y}_2}{2}$ and $\widehat{P}_\phi^3(1) := \widehat{P}_\phi(1|\hat{Y}_1, \hat{Y}_2) = 1 - (\hat{Y}_1 + \hat{Y}_2)/2$. Here, it is evident that $\widehat{P}_\phi(y|Y_{\pi(1)}, Y_{\pi(2)}) = \widehat{P}_\phi(y|Y_1, Y_2)$ satisfying Property 1. However, we observe that $\mathbb{E}(\widehat{P}_\phi^3(0) \mid \hat{Y}_1) = \mathbb{E}((\hat{Y}_1/2 + \hat{Y}_2/2) \mid \hat{Y}_1) = \hat{Y}_1/2 + 1/3$ is not equal to $\widehat{P}_\phi^2(0) = 1/3$. Thus, $\mathbb{E}(\widehat{P}_\phi^{t+1}(y) \mid \hat{Y}_{1:t-1}) \neq \widehat{P}_\phi^t(y)$, indicating that the sequence model is not exchangeable.

## D.3  Proof of Theorem 2

Recall that $y_{t+1:T} \sim \mathbb{P}(\cdot|y_{1:t})$ is the data generating process. Further, $\widehat{P}_\phi^O(y_{t+1:T}) \equiv \prod_{i=t+1}^T \widehat{P}_\phi(\hat{Y} = y_i|y_{1:t})$ is the one-step inference model and $\widehat{P}_\phi^M(y_{t+1:T}) \equiv \prod_{i=t+1}^T \widehat{P}_\phi(\hat{Y} = y_i|y_{1:i-1})$ is the multi-step inference model, where $\hat{Y}_i$ is generated autoregressively.

**Theorem 2:** Assuming $\widehat{P}_\phi = \mathbb{P}$, then the difference $\mathbb{E}\left(\log\left[\widehat{P}_\phi^M(y_{t+1:T})\right]\right) - \mathbb{E}\left(\log\left[\widehat{P}_\phi^O(y_{t+1:T})\right]\right)$ is equal to

$$\sum_{i=t+1}^T I(y_{t+1:i-1}; y_i|y_{1:t})$$

where $I(A; B|C)$ is the mutual information between $A$ and $B$ conditional on $C$ and expectation is w.r.t. $y_{1:T} \sim \mathbb{P}(\cdot)$.

**Proof**  As data is generated from $\mathbb{P}(\cdot)$. Therefore, under the assumption that $\widehat{P}_\phi = \mathbb{P}$, we have that

$$\mathbb{E}\left(\log\left[\widehat{P}_\phi^M(y_{t+1:T})\right]\right) - \mathbb{E}\left(\log\left[\widehat{P}_\phi^O(y_{t+1:T})\right]\right) =^{(a)} \mathbb{E}\left(\log\left[\prod_{i=t+1}^T \widehat{P}_\phi(\hat{Y} = y_i|y_{1:i-1})\right]\right)$$

$$- \mathbb{E}\left(\log\left[\prod_{i=t+1}^T \widehat{P}_\phi(\hat{Y} = y_i|y_{1:t})\right]\right)$$

$$=^{(b)} \mathbb{E}\left(\log\left[\prod_{i=t+1}^T \mathbb{P}(Y = y_i|y_{1:i-1})\right]\right)$$

$$- \mathbb{E}\left(\log\left[\prod_{i=t+1}^T \mathbb{P}(Y = y_i|y_{1:t})\right]\right)$$

$$= \mathbb{E}\left(\log\left[\frac{\prod_{i=t+1}^T \mathbb{P}(Y = y_i|y_{1:i-1})}{\prod_{i=t+1}^T \mathbb{P}(Y = y_i|y_{1:t})}\right]\right)$$

$$= \mathbb{E}\left( \sum_{i=t+1}^{T} \log\left[ \frac{\mathbb{P}(Y = y_i | y_{1:i-1})}{\mathbb{P}(Y = y_i | y_{1:t})} \right] \right)$$

$$= \sum_{i=t+1}^{T} \mathbb{E}\left( \log\left[ \frac{\mathbb{P}(Y = y_i | y_{1:i-1})}{\mathbb{P}(Y = y_i | y_{1:t})} \right] \right)$$

$$=^{(c)} \sum_{i=t+1}^{T} I(y_i, y_{t+1:i-1} | y_{1:t})$$

Here (a) follows from the definition of $\widehat{P}_\phi^M$, $\widehat{P}_\phi^O$. (b) follows from the assumption $\widehat{P}_\phi = \mathbb{P}$. (c) follows from the following identity

$$I(X;Y \mid Z) = \mathbb{E}\left[ \ln \frac{P(X,Y \mid Z)}{P(X \mid Z)\,P(Y \mid Z)} \right] = \mathbb{E}\left[ \ln \frac{P(X \mid Y,Z)}{P(X \mid Z)} \right] = \mathbb{E}\left[ \ln \frac{P(Y \mid X,Z)}{P(Y \mid Z)} \right].$$

Also note that it is equal to $\mathbb{E}_{y_{1:t} \sim \mathbb{P}}\left( D_{\mathrm{kl}}\left( \mathbb{P}(y_{t+1:T} | y_{1:t}) \| \prod_{i=t+1}^{T} \mathbb{P}(y_i | y_{1:t}) \right) \right)$

$\square$

### D.4 Proof of Example 1

**Proof**

Recall that, $Y_i = \theta + \varepsilon_i$, $\varepsilon_i \sim \mathcal{N}(0, \tau^2)$, $\theta \sim \mathcal{N}(\mu, \sigma^2)$.. We first estimate KL divergence

$$D_{\mathrm{KL}}\left( \mathbb{P}(y_{t+1:T} \mid y_{1:t}) \, \Big\| \, \prod_{i=t+1}^{T} \mathbb{P}(y_i \mid y_{1:t}) \right).$$

Because $Y_i \mid \theta \sim \mathcal{N}(\theta, \tau^2)$, therefore the posterior $\mathbb{P}(\theta \mid y_{1:t})$ is normal with mean $\mu_t = \frac{\frac{\mu}{\sigma^2} + \frac{1}{\tau^2} \sum_{i=1}^{t} y_i}{\frac{1}{\sigma^2} + \frac{t}{\tau^2}}$ and variance is $\sigma_t^2 = \left( \frac{1}{\sigma^2} + \frac{t}{\tau^2} \right)^{-1}$. Further conditional on $\theta$, the future $y_{t+1}, \ldots, y_T$ are i.i.d. $\mathcal{N}(\theta, \tau^2)$. Therefore we get, $y_{t+1:T} \mid y_{1:t} \sim \mathcal{N}\left( \mu_t \mathbf{1}, \ \tau^2 I + \sigma_t^2 \mathbf{1}\mathbf{1}^T \right)$, where $\mathbf{1}$ is the $(T-t)$-dimensional vector of all ones, and $I$ the $(T-t) \times (T-t)$ identity.

For the one-step inference $\prod_{i=t+1}^{T} \mathbb{P}(y_i \mid y_{1:t})$, each $y_i$ has distribution $y_i \mid y_{1:t} \sim \mathcal{N}(\mu_t, \tau^2 + \sigma_t^2)$. Denote this covariance by $\Sigma_Q = (\tau^2 + \sigma_t^2) I$.

Now, we want the KL divergence between two $(T-t)$ dimensional Gaussians with $P = \mathcal{N}(\mu_t \mathbf{1}, \Sigma_P)$ with $\Sigma_P = \tau^2 I + \sigma_t^2 \mathbf{1}\mathbf{1}^T$. and $Q = \mathcal{N}(\mu_t \mathbf{1}, \Sigma_Q)$ with $\Sigma_Q = (\tau^2 + \sigma_t^2) I$. For two $K$-dimensional Gaussians $\mathcal{N}(\mu_0, \Sigma_0)$ and $\mathcal{N}(\mu_1, \Sigma_1)$, the KL divergence is

$$D_{\mathrm{KL}}\big( \mathcal{N}(\mu_0, \Sigma_0) \, \| \, \mathcal{N}(\mu_1, \Sigma_1) \big) = \tfrac{1}{2}\left[ \mathrm{tr}\big( \Sigma_1^{-1} \Sigma_0 \big) + (\mu_1 - \mu_0)^T \Sigma_1^{-1} (\mu_1 - \mu_0) - K + \ln \frac{|\Sigma_1|}{|\Sigma_0|} \right].$$

Therefore we get -

$$D_{\mathrm{KL}}(P \| Q) = \tfrac{1}{2}\left[ K \ln\left( 1 + \frac{\sigma_t^2}{\tau^2} \right) - \ln\left( 1 + K \frac{\sigma_t^2}{\tau^2} \right) \right].$$

As the expression is independent of $y_{1:t}$, taking expectation over $y_{1:t}$ our final expression remains the same.

$\square$

## D.5 Generalization of Theorem 2 to the contextual setting

**Loss of information in one-step v/s multi-step:** Let $y_{t+1:T} \sim \mathbb{P}(\cdot|y_{1:t}, x_{1:T})$ be some data from true data generating process. Further, let $\widehat{P}^O_\phi(y_{t+1:T}, x_{t+1:T}) \equiv \prod_{i=t+1}^T \widehat{P}_\phi(\hat{Y}_{t+1} = y_i|y_{1:t}, x_{1:t}, x_i)$ and $\widehat{P}^M_\phi(y_{t+1:T}, x_{t+1:T}) \equiv \prod_{i=t+1}^T \widehat{P}_\phi(\hat{Y}_i = y_i|y_{1:i-1}, x_{1:i-1}, x_i)$.

**Theorem 4.** *Assume $\widehat{P}_\phi = \mathbb{P}$. The difference* $\mathbb{E}\left(\log\left[\widehat{P}^M_\phi(y_{t+1:T}, x_{t+1:T})\right] - \widehat{P}^O_\phi(y_{t+1:T}, x_{t+1:T})\right)$ *is*

$$\sum_{i=t+1}^T I(y_i; y_{t+1:i-1}|y_{1:t}, x_{1:i}) \tag{2}$$

*where expectation is $y_{1:T} \sim \mathbb{P}(\cdot|x_{1:T})$ and $x_{1:T} \overset{iid}{\sim} P_X$.*

### D.5.1 Proof of Theorem 4

Recall that $y_{t+1:T} \sim \mathbb{P}(\cdot|y_{1:t}, x_{1:T})$ is generated from the true data generating process. Further, $\widehat{P}^O_\phi(y_{t+1:T}, x_{t+1:T}) \equiv \prod_{i=t+1}^T \widehat{P}_\phi(\hat{Y} = y_i|y_{1:t}, x_{1:t}, x_i)$ and $\widehat{P}^M_\phi(y_{t+1:T}, x_{t+1:T}) \equiv \prod_{i=t+1}^T \widehat{P}_\phi(\hat{Y} = y_i|y_{1:i-1}, x_{1:i-1}, x_i)$.

**Theorem 4:** Assuming $\widehat{P}_\phi = \mathbb{P}$, then the difference $\mathbb{E}\left(\log\left[\widehat{P}^M_\phi(y_{t+1:T}, x_{t+1:T})\right]\right) - \mathbb{E}\left(\widehat{P}^O_\phi(y_{t+1:T}, x_{t+1:T})\right)$ is equal to

$$\sum_{i=t+1}^T I(y_i; y_{t+1:i-1}|y_{1:t}, x_{1:i})$$

where expectation is $y_{1:T} \sim \mathbb{P}(\cdot|x_{1:T})$ and $x_{1:T} \overset{iid}{\sim} P_X$.

**Proof**   We can follow exactly same procedure as in the proof of Theorem 2 for this proof.

$\square$

## D.6 Characterizing (2) for Bayesian linear regression and Gaussian processes

**Example 4. Bayesian Linear Regression -** Assuming $Y = \theta^T X + \epsilon$ where $\epsilon \sim N(0, \tau^2)$ and $\theta \sim N(\mu, \Sigma)$. Inputs $X$ are drawn i.i.d. from $P_X$. Let $\mathcal{D}_t := (x_{1:t}, y_{1:t})$ and the posterior $\theta|\mathcal{D}_t \sim N(\mu', \Sigma')$, then expression 2 is equal to

$$\frac{1}{2}\mathbb{E}_{\mathbf{X}, \Sigma'}\left[\log\frac{|\text{diag}(\mathbf{X}\Sigma'\mathbf{X}^T)|}{|\mathbf{X}\Sigma'\mathbf{X}^T|}\right]$$

where $\mathbf{X}$ is the matrix of $x_{t+1}, x_{t+2}\cdots, x_T$.

**Example 5. Gaussian Processes -** Assuming $Y = f(X) + \epsilon$ where $f \sim \mathcal{GP}(m, \mathcal{K})$, where $m(X)$ is mean and $\mathcal{K}(X, X')$ is the covariance. Additionally, Gaussian noise $N(0, \sigma^2)$ is added to the outputs. The input $X$ is drawn i.i.d. from $P_X$. Let $\mathcal{D}_t := (x_{1:t}, y_{1:t})$. Suppose, under multi-step inference $P(\mathbf{y}_{t+1:T} \mid \mathbf{y}_{1:t}, \mathbf{X}_{1:T}) = \mathcal{N}(\mathbf{y}_{t+1:T} \mid \boldsymbol{\mu}_P, \mathbf{K}_P)$. Further, under single-step inference $Q(\mathbf{y}_{t+1:T}) = \prod_{i=t+1}^T \mathcal{N}(y_i \mid \mu_i, \sigma_i^2) = \mathcal{N}(\mathbf{y}_{t+1:T} \mid \boldsymbol{\mu}_Q, \boldsymbol{\Sigma}_Q)$. Then, expression 2 is equal to

$$\frac{1}{2}\mathbb{E}_{\boldsymbol{\Sigma}_P, \boldsymbol{\Sigma}_Q, \mu_Q, \mu_P}\left[\log\left(\frac{\boldsymbol{\Sigma}_Q}{|\mathbf{K}_P|}\right) - (T - t) + \text{tr}\left(\mathbf{K}_P^{-1}\boldsymbol{\Sigma}_Q\right) + (\boldsymbol{\mu}_Q - \boldsymbol{\mu}_P)^\top \mathbf{K}_P^{-1}(\boldsymbol{\mu}_Q - \boldsymbol{\mu}_P)\right]$$

where $d = T - t$)

and the posterior $\theta|\mathcal{D}_t \sim N(\mu', \Sigma')$, then expression 2 is equal to

$$\frac{1}{2}\mathbb{E}_{\mathbf{X}, \Sigma'}\left[\log\frac{|\text{diag}(\mathbf{X}\Sigma'\mathbf{X}^T)|}{|\mathbf{X}\Sigma'\mathbf{X}^T|}\right]$$

where $\mathbf{X}$ is the matrix of $x_{t+1}, x_{t+2}\cdots, x_T$.

### D.6.1 Proof of Example 4 and 5

**Proof**

Recall that, under multi-step inference $P(\mathbf{y}_{t+1:T} \mid \mathbf{y}_{1:t}, \mathbf{X}_{1:T}) = \mathcal{N}(\mathbf{y}_{t+1:T} \mid \boldsymbol{\mu}_P, \mathbf{K}_P)$. Further, under single-step inference $Q(\mathbf{y}_{t+1:T}) = \prod_{i=t+1}^{T} \mathcal{N}(y_i \mid \mu_i, \sigma_i^2) = \mathcal{N}(\mathbf{y}_{t+1:T} \mid \boldsymbol{\mu}_Q, \boldsymbol{\Sigma}_Q)$. Now, KL Divergence between two multivariate Gaussian distributions: $\mathcal{N}(\mathbf{y} \mid \boldsymbol{\mu}_P, \mathbf{K}_P)$ and $Q = \mathcal{N}(\mathbf{y} \mid \boldsymbol{\mu}_Q, \boldsymbol{\Sigma}_Q)$ is give by

$$D_{\mathrm{kl}}(P\|Q) = \frac{1}{2}\left[\log\left(\frac{|\boldsymbol{\Sigma}_Q|}{|\mathbf{K}_P|}\right) - d + \mathrm{tr}\left(\mathbf{K}_P^{-1}\boldsymbol{\Sigma}_Q\right) + (\boldsymbol{\mu}_Q - \boldsymbol{\mu}_P)^\top \mathbf{K}_P^{-1}(\boldsymbol{\mu}_Q - \boldsymbol{\mu}_P)\right]$$

where: $d$ is the dimensionality of $\mathbf{y}$ (i.e., $d = T - t$); $|\cdot|$ denotes the determinant and $\mathrm{tr}(\cdot)$ denotes the trace of a matrix. Therefore, we get

$$D_{\mathrm{kl}}(P\|Q) = \frac{1}{2}\left[\log\left(\frac{\prod_{i=t+1}^{T}\sigma_i^2}{|\mathbf{K}_P|}\right) - (T-t) + \mathrm{tr}\left(\mathbf{K}_P^{-1}\boldsymbol{\Sigma}_Q\right) + (\boldsymbol{\mu}_Q - \boldsymbol{\mu}_P)^\top \mathbf{K}_P^{-1}(\boldsymbol{\mu}_Q - \boldsymbol{\mu}_P)\right]$$

In **Bayesian linear regression** the means remain the same, i.e., $\boldsymbol{\mu}_Q = \boldsymbol{\mu}_P$, further the expression reduces to the following:

$$D_{\mathrm{kl}}\left(\mathcal{N}(\mathbf{X}\mu', \mathbf{X}\Sigma'\mathbf{X}^T + \tau^2\mathbf{I})\|\mathcal{N}(\mathbf{X}\mu', \mathrm{diag}(\mathbf{X}\Sigma'\mathbf{X}^T) + \tau^2\mathbf{I})\right)$$

that is,

$$D_{\mathrm{kl}}(P\|Q) = \frac{1}{2}\left[\log\frac{|\mathbf{X}\Sigma'\mathbf{X}^T|}{|\mathrm{diag}(\mathbf{X}\Sigma'\mathbf{X}^T)|}\right]$$

Final expressions follows from proof of Theorem 2 in Section D.3.

$\square$

### D.7 Proof of Theorem 3

Recall that our setting was such that the reward of first arm is generated as $Y^{(1)} \sim N(\theta, \tau^2)$ with $\theta \sim N(\mu, \sigma^2)$. While the second arm has a constant reward $Y^{(2)} = 0$.

**Theorem 3:** There exists one-armed bandit scenarios in which Thompson sampling incurs $O(T)$ Bayesian regret if it relies solely on one-step predictions from autoregressive sequence models.

**Proof** Consider a bandit setting, where $\sigma = 0$ and $\mu < 0$. Implementing Thompson sampling using the one-step inference, does not differentiate between aleatoric and epistemic uncertainty. This means whenever we will sample $Y > 0$ we will choose the wrong arm to pull as $\mu$(mean reward from arm 1) $< 0$(mean reward from Arm 2). As $Y \sim N(\mu, \tau^2)$ therefore, $P(Y > 0) = \Phi\left(\frac{\mu}{\tau}\right)$. Hence, on an average we will suffer $\left(T\Phi\left(\frac{\mu}{\tau}\right)\right)$ regret over horizon $T$.

Similarly, when $\sigma = 0$ and $\mu > 0$, whenever we will sample $Y < 0$ we will choose the wrong arm to pull. Therefore in this case, it suffers $T\Phi\left(\frac{-\mu}{\tau}\right)$ Bayesian regret over horizon $T$. On the other hand, the multi-step inference (assuming it is done till $\infty$) will have 0 regret in these two case.

$\square$

## E  Additional Experiments

In this section, we present ablation studies to evaluate the performance of different architectures, specifically comparing conditionally permutation-invariant and standard causal masking schemes. In

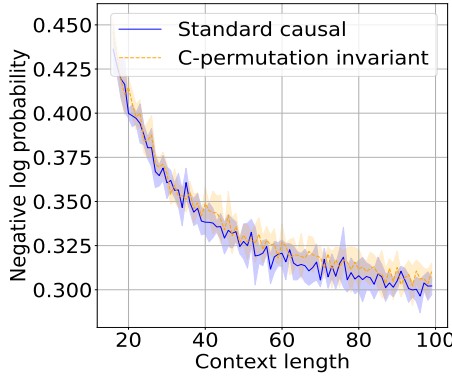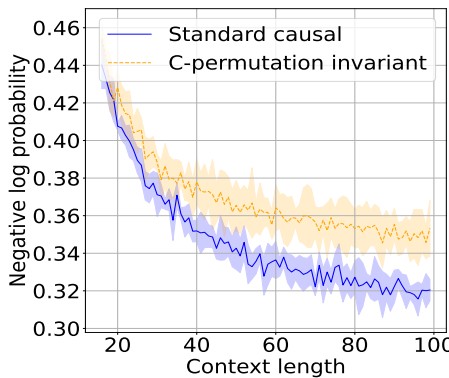

| **(a)** In-training horizon performance | **(b)** Out-of-training horizon performance |

Figure 11: **One-step log-loss:** Comparing the two architectures on one-step log-loss metric [Dimension: 1].

Section E.1, we provide the corresponding results from Section 4.3 using one-step log-loss as the evaluation metric. Section E.2 explores the effect of varying the input dimension $X$, while Section E.3 examines the impact of different noise levels in the observed output $Y$.

### E.1 One-step log-loss figures corresponding to Section 4.3

In Figures 11(a) and 11(b), we present the in-training horizon and out-of-training horizon performance for the corresponding settings shown in Figures 5(a) and 5(c). The results exhibit similar trends, where both masking schemes perform equally well in-distribution. However, beyond the training horizon, the causal masking approach appears to outperform the conditionally permutation-invariant masking.

### E.2 Ablations on dimensions

In this section, we present the results of our ablation study on the dimension $d$ of the context, where $X \sim U[-2, 2]^d$.

*In-training horizon performance:* Figure 12 shows the in-training horizon performance for both architectures across different dimensions. Our findings align with previous observations. Notably, for $d = 16$, the provided context appears insufficient to improve prediction log-loss.

*Training/Data efficiency:* Figure 13 illustrates the training and data efficiency across different dimensions.

*Out-of-training horizon performance:* We only analyze out-of-training efficieny for the dimension 4 in addition to dimension 1. As shown in Figure 14, the results remain consistent with our earlier findings.

*Multi-step v/s One-step inference:* Figure 15 present a comparison of multi-step and one-step inference across different dimensions.

### E.3 Ablations on noise

*In-training horizon performance:* Figures 16 illustrates the in-training horizon performance for both architectures under varying levels of observation noise. Our findings remain consistent with previous observations.

*Training/Data efficiency:* Figures 17 shows the training and data efficiency across different levels of observation noise. The results align with our earlier findings.

*Multi-step v/s One-step inference:* Figures 18 present a comparison of multi-step and one-step inference under different observation noise levels.

### E.4 Downstream performance of the two architectures

Figure 19 compares the performance of the conditionally permutation-invariant architecture and the standard causal architecture in an active learning setting. Our results indicate that the standard causal architecture outperforms the conditionally permutation-invariant architecture.

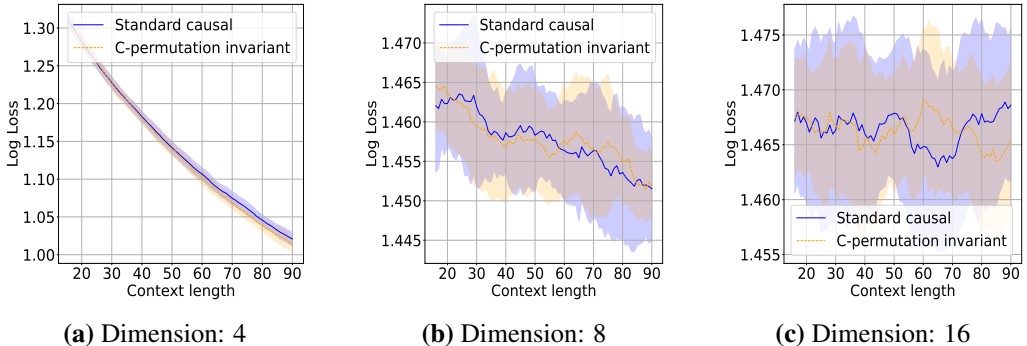

**(a)** Dimension: 4          **(b)** Dimension: 8          **(c)** Dimension: 16

Figure 12: **Ablation on dimension (In-training horizon performance):** Comparing two architectures [Training horizon: 100, Metric: Multi-step log-loss, Target length: 10].

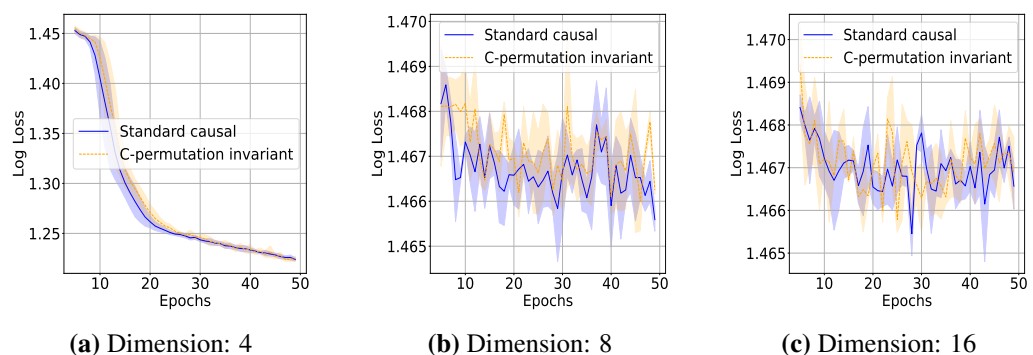

**(a)** Dimension: 4          **(b)** Dimension: 8          **(c)** Dimension: 16

Figure 13: **Ablation on dimension (Training/Data efficiency):** Comparing two architectures [Training horizon: 100, Metric: Multi-step log-loss, Target length: 10].

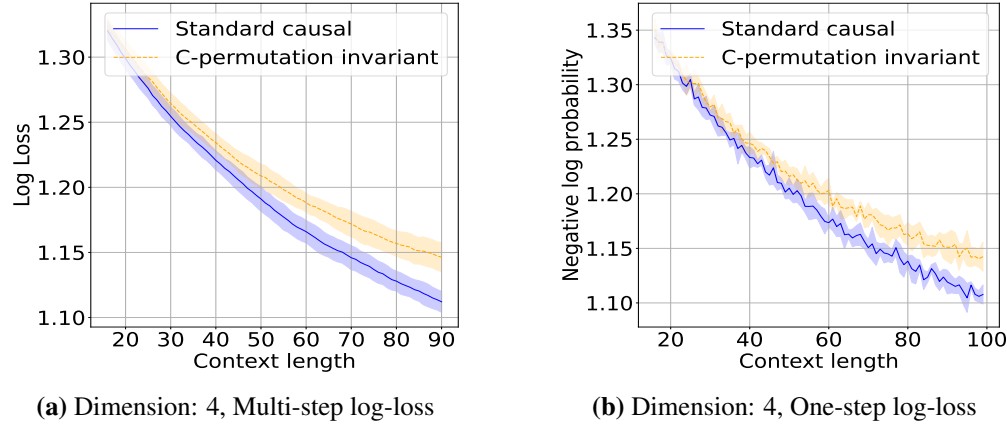

**(a)** Dimension: 4, Multi-step log-loss          **(b)** Dimension: 4, One-step log-loss

Figure 14: **Ablation on dimension (Out-of-training horizon performance):** Comparing two architectures [Training horizon: 15, Target length: 10].

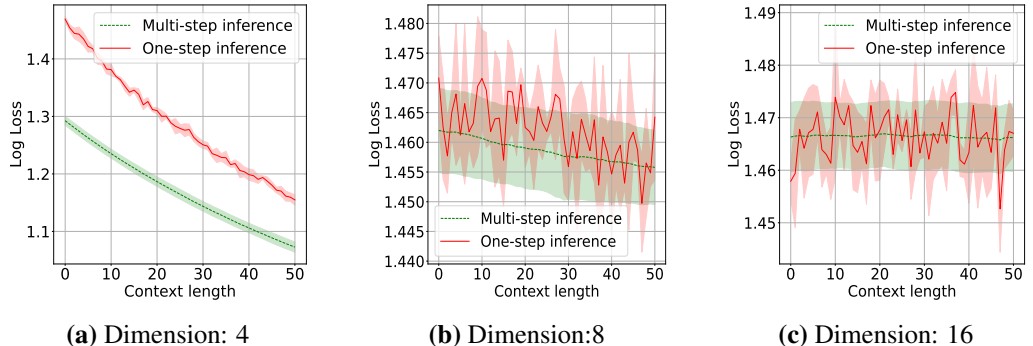

**(a)** Dimension: 4      **(b)** Dimension:8      **(c)** Dimension: 16

Figure 15: **Ablation on dimension (Uncertainty Quantification):** Comparing one-step inference and multi-step inference [Train horizon: 100, Metric: Multi-step log-loss, Target Length: 50].

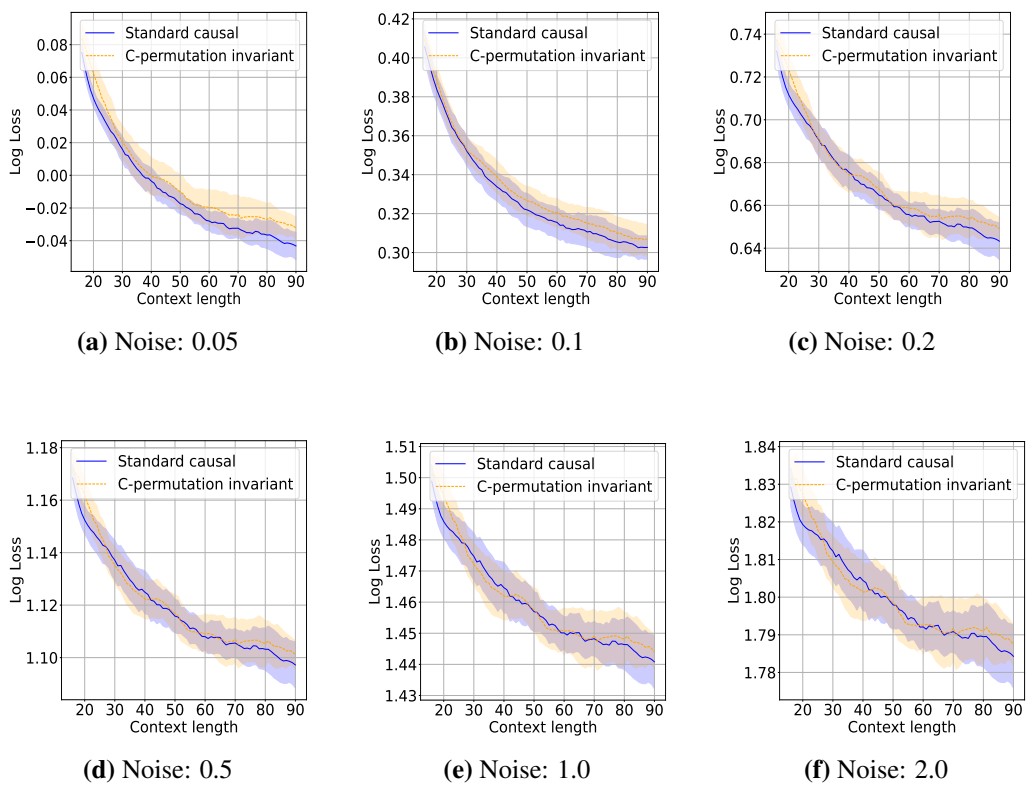

**(a)** Noise: 0.05      **(b)** Noise: 0.1      **(c)** Noise: 0.2

**(d)** Noise: 0.5      **(e)** Noise: 1.0      **(f)** Noise: 2.0

Figure 16: **Ablation on noise (In-training horizon performance):** Comparing two architectures [Training horizon: 100, Metric: Multi-step log-loss, Target length: 10].

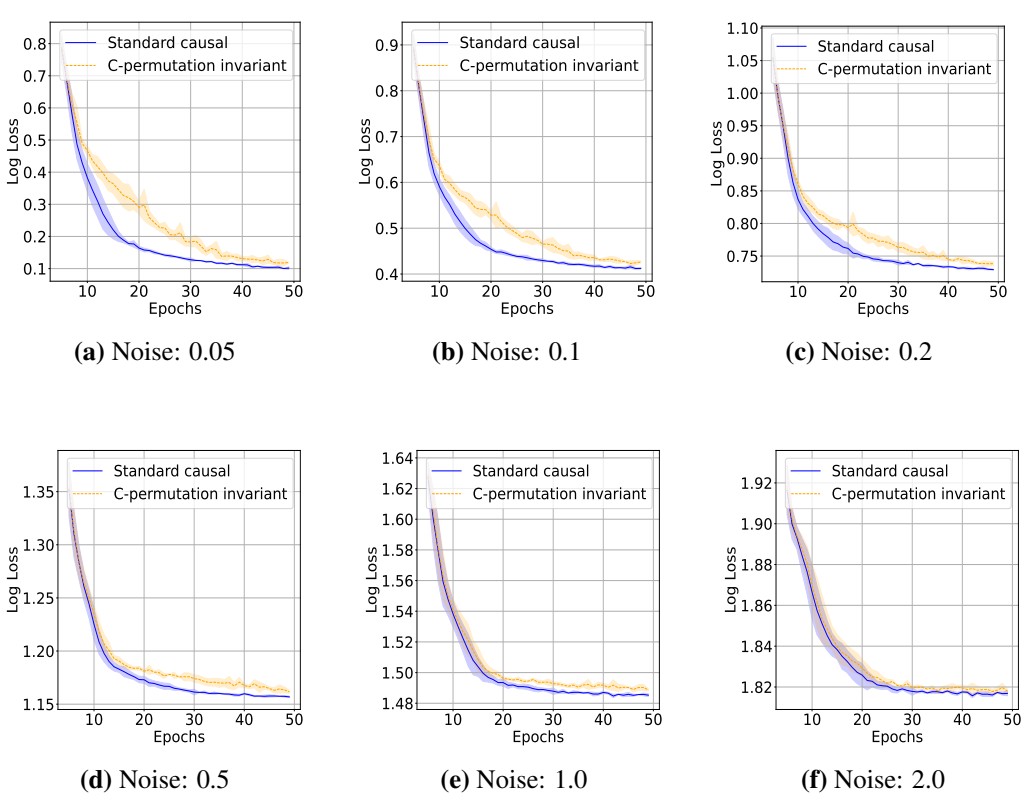

Figure 17: **Ablation on noise (Training/Data efficiency):** Comparing two architectures [Training horizon: 100, Metric: Multi-step log-loss, Target length: 10].

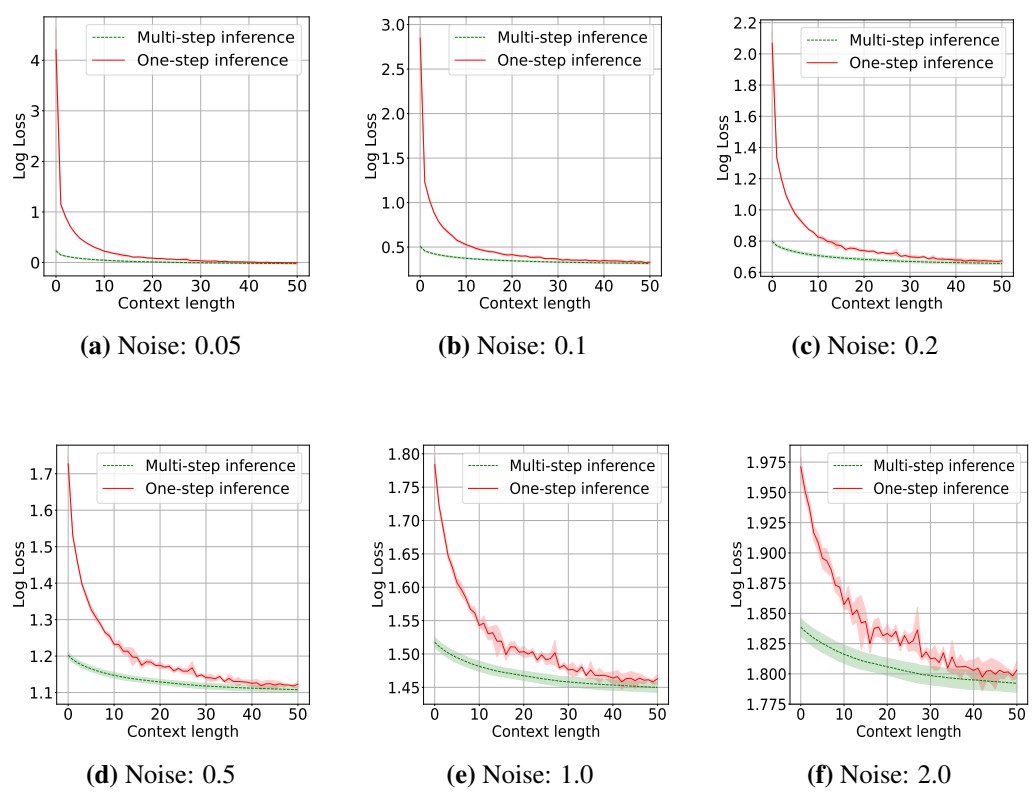

Figure 18: **Ablation on noise (Uncertainty Quantification):** Comparing One-step inference and multi-step inference [Train horizon: 100, Metric: Multi-step log-loss, Target Length: 50].

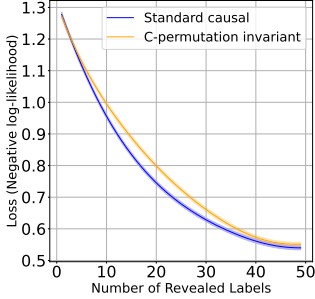

Figure 19: **Active Learning:** Comparing C-permutation invariant and standard causal architecture for the active learning setting.

