# OpenReview forum: "Architectural and Inferential Inductive Biases for Exchangeable Sequence Modeling"
_NeurIPS.cc/2025/Conference — NeurIPS 2025 poster_

### Official Review · Reviewer_JZjB · 2025-07-02

**Clarity:** 3
**Significance:** 3
**Originality:** 3
**Rating:** 5
**Confidence:** 2

**Summary:**

The paper tackles both the inferential and architectural inductive biases that matter when using Transformer sequence models on exchangeable data such as i.i.d. rows in a table.
On the inference side, it proves that the common one-step prediction heuristic systematically discards information.
On the architecture side, the authors show that recently proposed “exchangeable” masking schemes for Transformers is suboptimal.

**Questions:**

1. Is C-permutation invariant the same as conditional permutation invariant?
2. Based on my understanding, the proposed theory does not apply to most time series data due to exchangeable property, is that correct?
3. Can the authors provide some concrete example on how one apply either one-step or multi-step inference? A simple real world example would help a lot for understanding the paper.
4. Can the authors discuss how strong is the assumption $\hat{P_\phi} = \mathbb{P}$ ?

**Ethical Concerns:**

["NO or VERY MINOR ethics concerns only"]

**Final Justification:**

The authors have addressed my concerns, i've decided to remain my score.

**Limitations:**

Yes

**Quality:**

3

**Strengths And Weaknesses:**

## Strength
1. This paper tackles sequential modeling from a unique perspective. They show how the inference routine affects performance. I think this is a nice and meaningful perspective for the ML community.
2. The paper is well organized, it is easy to find information and follow.

## Weakness
1. Typo: line 223 (undertainty), appendix D.2 (exchangeablity), line 15 (infintely).
2. After proving existing masks insufficient, the paper did not provide a corresponding solution that satisfies both Property 1 and Property 2.

---

> ### Author Rebuttal · Authors · 2025-07-31
>
> We thank the reviewer for their thorough review and valuable feedback on our manuscript. Below, we address the specific questions raised. We will incorporate relevant portions of this discussion into the camera-ready version of the paper and thank the reviewer for their constructive feedback. Additionally, we have summarized the broader significance of this work and our main contributions in our response to the first reviewer (dFZM) and kindly request the reviewer to refer to that response as well.
>
>
> >**After proving existing masks insufficient, the paper did not provide a corresponding solution that satisfies both Property 1 and Property 2.**
>
> We acknowledge this limitation highlighted by the reviewer, which we also noted in our paper (lines 406-409). This represents a fundamental challenge in the field that our work has identified but not fully resolved. Specifically:
>
> 1. **Property 1 (conditional permutation invariance)** is relatively straightforward to enforce through architectural modifications such as Set Transformers, permutation-equivariant networks, or the masking schemes we analyzed.
>
> 2. **Property 2 (conditionally identically distributed/martingale property)** is significantly more challenging to impose architecturally. Traditional approaches typically assume parametric distributional forms, which limits representational power and undermines the flexibility that makes Transformers appealing for exchangeable sequence modeling.
>
> Our empirical findings suggest that standard causal masking may offer a reasonable intermediate solution for practitioners seeking to perform multi-step inference on exchangeable sequences, despite not enforcing permutation invariance. However, developing architectures that satisfy both properties while maintaining computational efficiency remains an important open research direction for future work.
>
> >**Can the authors discuss how strong is the assumption $\hat P_\phi =\mathbb P$ ?**
>
> This assumption is made primarily for theoretical clarity in Theorem 2 to isolate the fundamental information loss inherent in one-step versus multi-step inference. In practice, this assumption will not hold exactly due to finite data and model capacity limitations.
> When $\hat P_\phi \neq \mathbb P$, the information loss would include additional term proportional to KL divergence  $D_{KL}(\mathbb P \|\| \hat P_\phi)$. However, our key theoretical insight—that one-step inference systematically discards mutual information between future observations—remains valid regardless of model accuracy.
>
> >**Can the authors provide some concrete example on how one apply either one-step or multi-step inference? A simple real world example would help a lot for understanding the paper.**
>
> We provide detailed algorithmic descriptions for multi-step inference in active learning and multi-armed bandit settings in the appendices (Algorithms 2 and 3).
>
> To clarify this with a simple example, consider the coin scenario illustrated in Figure 2, which demonstrates how multi-step inference enables the distinction between epistemic and aleatoric uncertainty—a prerequisite for effective decision-making—while one-step inference fails to do so.
>
> **The Setup:** We have two coins (A and B) and must decide which to flip first to maximize expected rewards over multiple tosses.
>
> **One-step inference approach:**
>
> - Predicts only the immediate next outcome for each coin
>
> - Finds that both coins A and B have identical predictive uncertainty for their next toss
>
> - Cannot distinguish between the coins, leading to random selection
>
> **Multi-step inference approach:**
>
> - Simulates multiple future tosses for each coin to understand the uncertainty structure
>
> - Reveals a crucial difference:
>
>     -  **Coin B (epistemic uncertainty):** Multi-step simulation shows that after one toss, all future outcomes become deterministic. If the first toss is heads, all subsequent tosses will be heads; if tails, all will be tails. The uncertainty can be completely resolved with a single observation.
>
>    -  **Coin A (aleatoric uncertainty):** Multi-step simulation reveals that even after observing many tosses, each future toss remains uncertain due to the inherent randomness of a fair coin. The uncertainty is irreducible.
>
> - Multi-step inference correctly identifies that Coin B should be chosen first because its uncertainty can be resolved (providing valuable information), while Coin A's uncertainty cannot be reduced regardless of how many times it's observed
>
> This example illustrates the core advantage: multi-step inference captures the reducibility of uncertainty, which is essential for optimal sequential decision-making in applications like active learning and bandit problems.
>
>
> >**Is C-permutation invariant the same as conditional permutation invariant?**
>
> Yes, "C-permutation invariant" is our shorthand notation for "conditional permutation invariant." We will clarify this notation in the revision to avoid confusion.
>
>
> >**Based on my understanding, the proposed theory does not apply to most time series data due to exchangeable property, is that correct?**
>
> That is correct, our current theoretical framework specifically applies to exchangeable data and does not extend to time series data where temporal order matters. The exchangeability assumption is fundamental to our approach—time series data violates this assumption due to temporal dependencies.
>
>
> >**Typo: line 223 (undertainty), appendix D.2 (exchangeablity), line 15 (infintely).**
>
> Thank you for catching these errors. We will correct all typos in the camera-ready version.

---

### Official Review · Reviewer_Sm7a · 2025-07-02

**Clarity:** 4
**Significance:** 4
**Originality:** 3
**Rating:** 5
**Confidence:** 4

**Summary:**

This paper investigates the optimal inferential and architectural inductive biases for modeling exchangeable sequences with autoregressive models, particularly for downstream decision-making tasks. The authors make two primary contributions:
- First, they argue that the prevalent use of one-step inference is insufficient as it fails to distinguish between epistemic and aleatoric uncertainty. They advocate for multi-step autoregressive generation, demonstrating both theoretically and empirically that it provides superior uncertainty quantification, leading to better performance in active learning and bandit tasks.
- Second, the paper critically examines existing Transformer architectures designed to enforce exchangeability. The authors prove that these architectures only guarantee a weaker property of "conditional permutation invariance" and not full exchangeability. Through empirical evaluation, they show these specialized architectures are computationally inefficient and perform worse than a standard Transformer architecture.

**Questions:**

1. Given that standard causal models outperform specialized ones, what is the path forward for incorporating exchangeability as an inductive bias?
2. Multi-step inference is more computationally expensive. How should practitioners manage the trade-off between its superior performance and its higher computational cost in real-world applications?

**Ethical Concerns:**

["NO or VERY MINOR ethics concerns only"]

**Final Justification:**

Authors have agreed to add more empirical results in the camera-ready version. I would like to keep my "Accept" rating.

**Limitations:**

Yes

**Paper Formatting Concerns:**

well formatted

**Quality:**

4

**Strengths And Weaknesses:**

Strengths:
1. The paper makes a clear and significant argument for using multi-step inference in decision-making contexts, which is supported using both theory and experiments.
2. It provides a valuable and timely critique of prior architectural work, demonstrating that simpler causal models are more efficient and effective than complex "permutation-invariant" transformer designs that don't fully guarantee exchangeability

Weaknesses:
1. While the paper critiques existing architectures, it does not propose a novel, efficient architecture that can guarantee full exchangeability.
2. The experiments, while effective, are limited to synthetic data; validation on real-world datasets would further strengthen the claims.

---

> ### Author Rebuttal · Authors · 2025-07-31
>
> We thank the reviewer for their thorough review and valuable feedback on our manuscript. Below, we address the specific questions raised. We will incorporate relevant portions of this discussion into the camera-ready version of the paper and thank the reviewer for their constructive feedback. Additionally, we have summarized the broader significance of this work and our main contributions in our response to the first reviewer (dFZM) and kindly request the reviewer to refer to that response as well.
>
>
> >**While the paper critiques existing architectures, it does not propose a novel, efficient architecture that can guarantee full exchangeability.** and **Given that standard causal models outperform specialized ones, what is the path forward for incorporating exchangeability as an inductive bias?**
>
> Ensuring full exchangeability represents a central challenge highlighted by our work (lines 406-409). Our analysis demonstrates that merely enforcing Property 1 (conditional permutation invariance) is insufficient to guarantee full exchangeability. While Property 1 can be readily enforced through attention masking schemes (as demonstrated in prior work), Property 2 (the conditionally identically distributed/martingale property) poses a fundamental challenge.
>
> The difficulty arises because enforcing the martingale property typically necessitates imposing specific parametric constraints on the underlying distributions—constraints specifically designed to satisfy the martingale condition. However, such parametric restrictions significantly limit representational capacity, thereby undermining the core promise of Transformers and deep learning more broadly. This creates a fundamental tension between architectural expressivity and theoretical guarantees.
>
> Looking forward, our empirical findings suggest that **standard causal masking** may offer a reasonable intermediate solution for practitioners seeking to perform multi-step inference on exchangeable sequences, despite not enforcing strict permutation invariance. Additionally, we identify several promising research directions:
>
> 1. **Hybrid approaches:** Combining autoregressive models with architectural components that partially enforce exchangeability while maintaining computational efficiency
>
> 2. **Alternative architectures:** Exploring state-space models or Set Transformers that may offer better trade-offs between exchangeability guarantees and computational efficiency
>
> 3. **Regularization-based approaches:** Developing training objectives that encourage exchangeable behavior without hard architectural constraints
>
> 4. **Theoretical advances:** Characterizing when approximate exchangeability is sufficient for decision-making tasks
>
> Our finding that standard causal masking outperforms specialized architectures suggests that the inductive bias of exchangeability may be better incorporated through training procedures or regularization rather than architectural constraints.
>
>
>
> > **The experiments, while effective, are limited to synthetic data; validation on real-world datasets would further strengthen the claims.**
>
> We appreciate this feedback. While our main paper focuses on controlled synthetic experiments to clearly demonstrate the theoretical principles, we will include an additional experiment with real-data example in the camera-ready version, showcasing the benefits of multi-step inference over the one-step inference.
>
>
>
> > **How should practitioners manage the trade-off between its superior performance and its higher computational cost in real-world applications?**
>
> The computational overhead of multi-step inference scales linearly with the lookahead horizon (multi-step inference window) $H$, requiring $H$ times more computation than single-step inference. However, practitioners can effectively manage this computational trade-off through:
>
> **Optimal Horizon Selection:**  For any fixed computational budget, an optimal trade-off exists between lookahead horizon length and the number of generated trajectories for uncertainty estimation. This optimum is fundamentally determined by dataset characteristics, including:
> - Signal-to-noise ratio
> - Relative magnitudes of epistemic versus aleatoric uncertainty
> - Data dimensionality and complexity
>
> **Practical Guidelines:** We recommend adopting a validation-based approach whereby practitioners incrementally increase horizon length until the empirical mean converges and variance estimates stabilize. Subsequently, the horizon should be balanced against trajectory count based on available computational constraints.
>
> Importantly, in many decision-making applications (e.g., active learning, clinical trials), the cost of acquiring new data substantially exceeds computational costs, thereby justifying the additional computational overhead.
>
> **Theoretical Characterization:** Developing theoretical frameworks to predict optimal horizon lengths given specific dataset characteristics remains an important open problem and represents a promising direction for future research.

---

### Official Review · Reviewer_UKPT · 2025-07-02

**Clarity:** 3
**Significance:** 2
**Originality:** 3
**Rating:** 4
**Confidence:** 2

**Summary:**

This paper reveals a potential flaw of using Transformers trained with one-instance generation loss for posterior inference. The work shows that the current approaches to building and using sequence models for exchangeable data on regression tasks are often theoretically unsound and practically suboptimal.

**Questions:**

(Q1) The distinction between epistemic and aleatoric uncertainty is a key justification for multi-step inference. However, this approach introduces practical challenges. Could you provide an analysis of the computational overhead of this method? How does the inference time scale with the "imagination horizon" (the length of generated sequences) and the size of the model? Following this, how should one determine the optimal imagination horizon? Is there a risk that a poorly chosen horizon could lead to noisy or biased uncertainty estimates, and how sensitive are the downstream task improvements to this choice?

(Q2) Following (W3), the experiments mainly demonstrate your points on synthetic data. How do you expect these findings to translate to large-scale, noisy, real-world tabular datasets where features may have complex dependencies? Would the performance gap between single-step and multi-step inference become more or less pronounced in such settings?

(Q3) The paper convincingly demonstrates the limitations of applying autoregressive models to exchangeable data. This raises a fundamental question: what is the rationale for focusing on adapting Transformers for this task, rather than exploring architectures inherently designed for permutation-invariance, such as Deep Sets, or other graph-based neural networks?

(Q4) What would be equivalent problem/strategy when number of steps becomes large?

**Ethical Concerns:**

["NO or VERY MINOR ethics concerns only"]

**Final Justification:**

Thanks for the addressing my comments and answering my questions. It makes sense to limit the scope to Transformers. I still believe experiments on real-world data/applications would increase the value of the work.

**Limitations:**

yes

**Quality:**

3

**Strengths And Weaknesses:**

Strength

(S1) The paper's primary strength is its formal and clear-headed analysis. The distinction drawn between "conditional permutation invariance" and true "exchangeability" is a particularly insightful and valuable contribution that clarifies a subtle but critical misunderstanding in prior work.

(S2) The paper provides compelling evidence that specialized attention mechanisms, despite being designed to handle exchangeability, can fail in practice. This finding serves as an important and actionable "negative result" for researchers, cautioning against the pursuit of complex architectural fixes that may not be theoretically sound.

(S3) By demonstrating the fundamental mismatch between autoregressive models and exchangeable data, the work implicitly questions the community's default choice of Transformers for this problem class. It forces a necessary and critical re-evaluation of whether adapting ordered models is a more fruitful path than developing architectures inherently suited to permutation-invariance.

Weakness

(W1) The paper's motivation is well-grounded in showing that current Transformer-based approaches are flawed. However, it does not sufficiently justify why the community should continue to focus on adapting Transformers for this task at all, rather than exploring more natural alternatives like Deep Sets or other architectures explicitly designed for permutation-invariant data.

(W2) The paper's critique, while sharp, is focused on a small and specific set of recent works. To broaden its impact and appeal to a general audience, the paper would benefit from contextualizing its findings within a wider survey of methods for modeling exchangeable data, thereby making a stronger case for the novelty and significance of its specific contributions.

(W3) While using Gaussian Processes as a case study is a reasonable starting point, the paper's claims about modeling general tabular data would be substantially more convincing with a broader empirical evaluation. Demonstrating the identified issues and the benefits of multi-step inference on a variety of well-known, real-world tabular datasets is essential to validate the practical relevance of the findings.

---

> ### Author Rebuttal · Authors · 2025-07-31
>
> We thank the reviewer for their thorough review and valuable feedback on our manuscript. Below, we address the specific questions raised. We will incorporate relevant portions of this discussion into the camera-ready version of the paper and thank the reviewer for their constructive feedback. Additionally, we have summarized the broader significance of this work and our main contributions in our response to the first reviewer (dFZM) and kindly request the reviewer to refer to that response as well.
>
>
> >**W1: Why Focus on Transformers Rather Than Natural Alternatives?** and **Q3: Rationale for Transformer Focus**
>
> Thanks for raising an excellent question about exploring alternatives like Deep Sets. We focus on Transformers for three key reasons:
>
> 1. **Scalability and Practical Impact:** Transformers have demonstrated unprecedented scalability and are the current SOTA for sequence modeling. Given their widespread adoption in practice, understanding their inductive biases in the context of  exchangeable data has immediate practical implications.
>
> 2. **Meta-learning Capabilities:** Unlike traditional methods that typically model single datasets with parametric distributions, Transformer-based sequence models can meta-learn across multiple datasets simultaneously.
>
> 3. **Foundation for Future Work:** Our theoretical analysis establishes fundamental principles for modeling exchangeable data that transcend specific architectural choices. These findings provide critical guidance for future research, particularly in understanding: (1) the trade-offs between multi-step and one-step inference approaches, and (2) the distinction between conditional permutation invariance and the martingale property.
>
> We agree that exploring inherently permutation-invariant architectures (e.g., State Space Models, Set Transformers, Deep Sets) represents important future work, and our findings will help inform these research directions as well. We will add this discussion to clarify our scope.
>
> >**W2: Broader Context and Survey**
>
> We thank the reviewer for this valuable suggestion. We will expand our related work to better contextualize our contributions within the broader uncertainty quantification landscape.
>
> **The Significance of This Work:** Despite decades of research, the ML community still faces fundamental challenges in uncertainty quantification. Most existing approaches—Bayesian neural networks, ensemble methods, variational inference—fall outside ML's core paradigm of optimizing loss functions and benchmarking. They require hand-crafted priors, complex approximations, or computationally prohibitive ensembles that don't scale to modern architectures. This methodological gap creates a fundamental tension: while representation learning drives modern ML's success, most principled UQ methods cannot effectively integrate with learned representations
>
> This scalability challenge directly impacts deployment in downstream decision-making tasks. Consider Thompson sampling, a cornerstone algorithm for sequential decision-making. Traditional implementations rely on closed-form posterior updates and hand-crafted priors, making them incompatible with the learned representations that power modern ML systems. This incompatibility has created a critical bottleneck that limits practical UQ deployment.
>
> **Autoregressive Sequence Modeling as a Paradigm Shift:** What makes this approach exciting is that it brings uncertainty quantification back into the realm of what ML does well. Through De Finetti's predictive view, we can directly model observables using standard optimization machinery, enabling principled uncertainty quantification without additional hyperparameters or architectural complexity.
>
> **Critical Gaps We Address:** However, this promising direction has important subtleties the community has overlooked: (1) one-step inference cannot distinguish epistemic from aleatoric uncertainty, and (2) specialized "exchangeable" architectures are based on informal intuitions rather than rigorous analysis. We address these subtleties in the paper leading to stronger foundation, and also demonstrate their effectiveness in downstream tasks.
>
> In addition, we will expand our survey to include:
>
> - Classical Bayesian approaches and their scalability limitations
> - Non-parametric methods (Dirichlet Processs)
> - Neural approaches (Deep Sets, Set Transformers, Neural Processes)
> - The positioning of autoregressive sequence modeling within this landscape
>
> This contextualization will clarify why our theoretical contributions—distinguishing conditional permutation invariance from true exchangeability, and demonstrating the fundamental importance of multi-step inference—represent critical advances for making uncertainty quantification practical in modern ML systems.
>
>
> >**W3: Real-World Validation** and **Q2: Real-World Dataset Performance**
>
> Thank you for the helpful feedback. In the camera-ready version, we will include an additional experiment on a real-data example to illustrate the benefits of multi-step inference compared to one-step inference in the active learning setting.
>
> Meanwhile, we note that the performance gap in real settings depends on the following dataset characteristics:
>
> - **Signal-to-noise ratio:** Higher noise requires longer horizons to distinguish epistemic from aleatoric uncertainty. For a fixed horizon, the gap between multi-step and single-step inference diminishes with increased noise levels.
>
> - **Epistemic vs. aleatoric ratio:** Higher epistemic uncertainty leads to larger performance gaps between multi-step and single-step inference.
>
> - **Dimension and heterogeneity:** Greater noise heterogeneity increases the performance gap between multi-step and single-step inference. Higher dimensionality may also amplify this gap.
>
>
> >**Q1: Computational Overhead and Horizon Selection**
>
> **Computational Complexity:** Multi-step inference scales as O(H×B×T²) where H is the imagination horizon (multi-step inference window), B is the number of trajectories, and T is the sequence length.  This complexity is H times greater than one-step inference, which requires O(B×T²) operations.
>
> While longer horizons generally yield better performance, for a fixed computational budget, there exists a fundamental trade-off between horizon length and the number of trajectories used for uncertainty estimation.
>
> **Optimal Horizon Selection:** The optimal horizon depends on dataset characteristics:
> - **Signal-to-noise ratio:** Higher noise requires longer horizons to distinguish epistemic from aleatoric uncertainty
> - **Epistemic vs. aleatoric ratio:** When aleatoric uncertainty dominates, the performance gap between inference methods diminishes, and requires longer horizons.
> - **Dimension and heterogeneity:** Complex dependencies may require longer horizons
>
> Theoretically characterizing this dependence represents an interesting avenue for future research.
>
> **Practical Guidelines:** We recommend a validation-based approach - incrementally increase horizon length until the empirical mean converges and variance estimates stabilize, then balance horizon against trajectory count based on computational constraints.
>
> >**Q4: Large Step Regime**
>
> For very long horizons, we might employ a heuristic approach of batching along with a diversity criteria.
>
> - **Batching:** Sampling multiple points based on uncertainty estimates from single step.
> - **Diversity Criteria:** Introduces diversity between sampling points.

---

> > ### Comment · Reviewer_UKPT · 2025-08-05
> >
> > Thanks for addressing my comments. I would appreciate if the authors can demonstrate the effectiveness through experiments on real-world data/applications.

---

### Official Review · Reviewer_dFZM · 2025-07-05

**Clarity:** 2
**Significance:** 2
**Originality:** 2
**Rating:** 3
**Confidence:** 1

**Summary:**

Autoregressive models such as Transformer-based model architectures are increasingly used for exchangeable sequence modeling, but common approaches rely on **one-step inference**, which fails to distinguish between epistemic uncertainty and aleatoric uncertainty.

This limitation negatively impacts decision-making tasks.

The paper did multiple analyses to show that
- multi-step inference in training is essential for reliable uncertainty quantification and downstream task performance;
- existing architectural approaches exchangeability are flawed and computationally expensive without providing benefits;
- designing transformer architectures that achieve full exchangeability while remaining computationally efficient remains an open challenge.

**Questions:**

n/a

**Ethical Concerns:**

["NO or VERY MINOR ethics concerns only"]

**Final Justification:**

I keep my ratings and have explained in more details in the response to the authors.

**Quality:**

2

**Strengths And Weaknesses:**

## Reviewer Expertise Disclaimer
This work falls outside my core area of expertise. While I have some familiarity with sequence modeling and uncertainty quantification, I encourage the area chairs to weigh this review accordingly, especially regarding technical novelty and empirical depth.

## Strengths
1. **Theoretical and Empirical Analysis**: The paper provides a thorough theoretical critique, in the context of uncertainty quantification, of one-step inference and supports its arguments with empirical results demonstrating the advantages of multi-step inference in specific tasks.
2. **Architectural Insights**: It offers a detailed examination of transformer architectures concerning **exchangeability**, contributing to the understanding of their limitations and guiding future architectural designs.

## Limitations and Concerns
1. **Limited scope of novelty**: The paper's core critique seems novel but only in a narrow context. Broad limitations of single-step inference and standard Transformer architectures have been explored in broader reasoning and planning contexts (e.g., Shojaee et al., 2025; Valmeekam et al., 2023; Creswell et al., 2022). While the paper’s domain (exchangeable sequences) is more narrow, the broader concern has precedent.
2. **Uncertainty quantification is conceptually important but underused in practice, especially in sequential decision making**: While uncertainty quantification is theoretically valuable, the claim that it is a prerequisite for intelligent agents is overstated. While uncertainty quantification is indeed valuable for certain types of decision-making, it is not a universally accepted prerequisite for intelligent behavior. Classical control systems, expert systems, and many successful RL algorithms function without explicit UQ mechanisms, despite the presence of partial observability and noisy environments.
3. **Limited generality of the scope**: Although the paper frames its contributions in terms of “decision-making,” the empirical evaluations are limited to multi-armed bandits and active learning, and especially in exchangeable settings. These are important, but relatively constrained, problem settings.

Combined, the scope of novelty and impact of the paper seems relatively limited for a NeurIPS publication.

## Recommendation
Due to my distance from the specific subdomain, I do not make a definitive accept/reject recommendation. I encourage a review from a domain expert who can more fully evaluate the novelty and technical depth of the paper.

---

> ### Author Rebuttal · Authors · 2025-07-31
>
> ## **General response for Reviewers/AC/SAC**
>
> We thank the reviewers for their positive and constructive feedback. We are glad that reviewers appreciate the clarity and significance of our theoretical analysis on exchangeability and multi-step inference vs. one-step inference (dFZM, UKPT, Sm7a), and recognize the value and timeliness of our empirical result on failure of architectural inductive biases (dFZM, UKPT, Sm7a, JZjB). At the same time, we acknowledge the valid concerns raised about specific aspects of our contribution. In light of these, we emphasize the broader significance of our contributions and contextualize it further.
>
> The machine learning community faces a fundamental challenge in uncertainty quantification (UQ). Modern ML systems are powered by representation learning, excelled through minimizing loss functions and benchmarking on standard datasets. Most existing UQ approaches fall outside this paradigm—requiring hand-crafted priors, complex variational approximations, or ensemble methods that don't scale to modern ML architectures.
>
> This methodological gap creates significant inefficiencies in downstream decision-making tasks. For example, Thompson sampling—a popular algorithm for active exploration—typically relies on hand-crafted priors and closed-form posterior updates. This limits practical deployment of principled decision-making algorithms in modern ML systems.
>
> **Autoregressive Sequence Modeling: A Paradigm Shift**
>
> What is exciting about autoregressive sequence modeling for exchangeable sequences is that it brings UQ back into ML's comfort zone. Instead of wrestling with intractable posteriors or hand-crafted priors, we directly model observables through De Finetti's predictive view. This approach:
> - Leverages the same optimization machinery (gradient descent on likelihood) that powers modern ML.
> - Enables direct benchmarking through standard metrics (log-likelihood, downstream task performance).
> - Scales naturally to large datasets.
> - Provides principled UQ without additional hyperparameters.
>
> This approach represents a fundamental shift in how we do UQ, making it compatible with tools and practices that have made modern ML successful. However, this promising direction involves important subtleties concerning inferential and architectural inductive biases that the community has largely overlooked. A central contribution of our work is the systematic identification and clarification of these subtleties both theoretically and empirically.
>
> **Summary of Contributions**
>
> Our work addresses a fundamental gap in autoregressive sequence modeling for exchangeable sequences, making three key contributions:
> 1. Analysis of One-Step vs Multi-Step Inference (also highlighted by dFZM, UKPT and JZjB)
>     - We prove that one-step inference fundamentally loses mutual information (Theorem 2), leading to degraded UQ.
>     - We demonstrate that this limitation causes one-step inference to suffer $O(T)$ regret in bandits setting (Theorem 3).
>      - Empirically, we show multi-step inference achieves up to 60% better performance in bandits and requires 10× fewer samples in active learning.
> 2. Formal Characterization of Architectural Inductive Biases (also highlighted by dFZM, UKPT, Sm7A)
>      - We introduce conditional permutation invariance (Property 1) to formally characterize what existing Transformer architectures actually enforce.
>      - We prove that conditional permutation invariance alone is insufficient for exchangeability, as it fails to guarantee the conditionally identically distributed (c.i.d.) property (Property 2).
>      - This corrects misconceptions in prior work (Müller et al., 2022; Nguyen & Grover, 2022; Ye & Namkoong, 2024) that implicitly assumed their architectures achieved full exchangeability.
> 3. Computational and Performance Analysis of Architectural Choices (also highlighted by dFZM, UKPT, Sm7A)
>     - Standard causal architectures outperform specialized "exchangeable" architectures by ~10\% on out-of-training horizons with 20% better training and data efficiency.
>     - Additionally, enforcing conditional permutation invariance introduces significant computational overhead at inference ($O(T^3)$ without KV caching benefits).
>
> **Summary of responses to other concerns raised by reviewers**
> 1. Reviewers (Sm7a, JZjB) point out that we do not propose a new architecture to address the limitations we identified. We acknowledged this limitation in our paper (lines 406-409). This represents a fundamental challenge in the field that our work has identified but not fully resolved (see details in response to reviewers Sm7a, JZjB). Our empirical findings suggest that standard causal masking may offer a reasonable intermediate solution. However, developing architectures that satisfy both properties while maintaining computational efficiency remains an important open research direction.
> 2. Reviewers (UKPT, Sm7a) raised concerns about computational overhead for multi-step inference in practice. In many practical settings where our approach would be useful, the benefits of better decision-making far outweigh the computational cost. For example, in clinical settings, the cost of data acquisition far exceeds computational costs. Additionally, one can determine an optimal lookahead horizon to prevent wasteful use of computational resources. Exploring this optimal horizon represents an interesting future direction—we also discuss the factors that impact the lookahead horizon with details in response to reviewers UKPT, Sm7a.
> 3. Based on feedback from reviewers (UKPT, Sm7a), we will include a real-data experiment in the camera-ready version demonstrating the benefits of multi-step inference over one-step inference.
>
> ## **Response to Reviewer dFZM**
>
> We thank the reviewer for their thorough review and valuable feedback on our manuscript. Below, we address the specific questions raised.
>
> >**Uncertainty quantification (UQ) is ... underused in practice**
>
> We agree that UQ remains underutilized in practice despite its theoretical importance. **Our work directly aims to address this gap.**
>
> As the reviewer notes, the theoretical need for reliable UQ in efficient exploration has been well-established (Osband et al., 2016; Charpentier et al., 2022; Deisenroth et al., 2011), yet practical deployment remains limited to specific domains like online recommendation systems (Li et al., 2019), clinical decision-making (Lopez et al., 2025), and hyperparameter optimization (Golovin et al., 2017).
>
> **The core issue is scalability.** Modern ML systems are powered by representation learning, excelled through minimizing loss functions and benchmarking on standard datasets. Most existing UQ approaches fall outside this paradigm—requiring hand-crafted priors, complex variational approximations, or ensemble methods that don't scale to modern ML architectures. This creates fundamental tension: while representation learning drives modern ML's success, most principled UQ methods cannot effectively integrate with learned representations.
>
> This scalability challenge directly impacts deployment in downstream decision-making tasks. Consider Thompson sampling, a cornerstone algorithm for sequential decision-making. Traditional implementations rely on closed-form posterior updates and hand-crafted priors, making them incompatible with learned representations that power modern ML systems. This incompatibility creates a critical bottleneck that limits practical UQ deployment.
>
> **Autoregressive sequence modeling for exchangeable sequences offers a solution**—By directly modeling observables through De Finetti's predictive view, it brings UQ back into ML's comfort zone—no intractable posteriors or hand-crafted priors required. This approach is inherently scalable and compatible with modern deep learning paradigms.
>
> However, this promising direction involves important subtleties that the community has largely overlooked, which we address in this paper:
> - **Inferential inductive biases:** The critical distinction between one-step and multi-step inference.
> - **Architectural inductive biases:** Proper enforcement of exchangeability and its empirical implications.
>
> We demonstrate the importance of these subtleties by focusing on data-acquisition tasks where agents must decide what data to collect next. In these settings, separating epistemic from aleatoric uncertainty is necessary for efficient performance, as shown by our theoretical results (Theorems 2-3) and empirical validation (Section 3).
>
> >**Limited scope of novelty** and **Limited generality of the scope**
>
> We thank the reviewer for the references to broader planning literature. While those works demonstrate advantages of multi-step planning in general reasoning tasks, our contribution is fundamentally different:
>
> 1. **Theoretical characterization:** We provide a formal analysis of why one-step inference fails in downstream decision-making tasks as compared to multi-step inference, quantifying the information loss (Theorem 2) and downstream impact (Theorem 3).
>
> 2. **Exchangeable data focus:** Our work targets sequence models meta-trained on tabular/exchangeable datasets—a rapidly growing area with applications in recommendation systems, clinical trials, and adaptive experimentation. This differs from the general reasoning contexts addressed in above mentioned works.
>
> 3. **Architectural insights:** We identify and formalize a critical gap in existing "exchangeable" architectures—they enforce only conditional permutation invariance, not full exchangeability, while being computationally expensive without performance benefits.
>
> Multi-armed bandits and active learning are canonical examples of decision-making on exchangeable data, providing analytical tractability while covering important practical applications. To our knowledge, both the theoretical analysis of multi-step vs. one-step inference and the architectural results for exchangeable sequence modeling are novel contributions.

---

> > ### Comment · Reviewer_dFZM · 2025-08-06
> >
> > I appreciate the authors for the responses.
> >
> > However, I don't think that the responses directly addressed my feedback.
> >
> > For example: my feedback for W2 is
> >
> > >Uncertainty quantification is conceptually important but underused in practice, especially in sequential decision making: While uncertainty quantification is theoretically valuable, the claim that it is a prerequisite for intelligent agents is overstated. While uncertainty quantification is indeed valuable for certain types of decision-making, it is not a universally accepted prerequisite for intelligent behavior. Classical control systems, expert systems, and many successful RL algorithms function without explicit UQ mechanisms, despite the presence of partial observability and noisy environments.
> >
> > The response simply didn't address these points.
> >
> > I also strongly disagree that scalability (assumed to be computational scalability, not generalization) is the reason for the lack of adoption of the existing UQ framework. For example, ensemble only increases the computational cost by a constant factor, while transformers increase the computational costs, compared to the previous practice, from linear to quadratic. Yet, the latter is widely used. And I don't agree that the paper offers a solution in this regard.
> >
> > Therefore, I will keep my review and rating as they are.

---

> > > ### Author Response · Authors · 2025-08-08
> > >
> > > Thank you for your response. We are sorry for any misunderstanding regarding your earlier point, and we agree that certain types of intelligent behavior may not require uncertainty quantification (UQ).
> > >
> > > We do not claim that explicit UQ is a prerequisite for intelligent behavior in general. Many effective systems (e.g., classical control, some RL variants) operate successfully without UQ. However, our claim is scoped to decisions based on exchangeable sequences where the decision policy is driven by posterior/predictive uncertainty (e.g., Thompson sampling, uncertainty sampling). In that setting, failing to separate epistemic from aleatoric uncertainty can be demonstrably harmful. For exchangeable data, UQ—whether explicit or implicit—already underlies numerous applications including recommendation systems, adaptive experimentation, hyperparameter optimization, clinical decision-making, and Bayesian optimization. In our paper, we illustrate this in the context of data acquisition tasks.
> > >
> > > Regarding “scalability”, we apologize for our lack of clarity. Our point is that Transformers are already being applied to meta-learning on tabular datasets, and the sequence-modeling view of exchangeable sequences enables natural incorporation of UQ in such models. This allows us to exploit the scalability (representational capacity, and generalization abilities) of Transformers for UQ as well. Our central argument is that, when applying sequence models to exchangeable data, one-step inference is insufficient for reliable UQ; multi-step inference is essential.
> > >
> > > We will revise the paper to clarify these points and make our scope more explicit.

---

### Decision · Program_Chairs · 2025-09-17

**Decision:**

Accept (poster)

**Comment:**

The paper proposes to revisit decision making along the following lines:
*  decision making, as opposed to mainstream classification, must consider uncertainty quantification (UQ, under-used in ML practice) and thus distinguish among epistemic and aleatoric uncertainty;
* this distinction requires to consider multiple-step inference (only enabling to reduce the epistemic uncertainty) instead of single-step inference.
* the literature handles exchangeable sequences (sequential events are independent conditionally to the distribution) through specialized masking strategies;
* still, "the invariance properties enforced by such masking schemes don’t necessarily guarantee valid probabilistic inference".

Basically, the paper introduces the property of conditional permutation invariance; shows that this property is enforced by transformer architectures, and that this property does not guarantee exchangeability, entailing a loss of performance.

Rev UPKT appreciates the clear-headed analysis. They suggest that broader experiments are essential to appreciate whether the benefits are worth the extra-cost of the approach, and that the critique should not focus on the only transformer architectures. Authors see their main contributions as i) distinguishing conditional permutation invariance from true exchangeability; ii) demonstrating the fundamental importance of multi-step inference, and state that these represent critical advances for making uncertainty quantification practical in modern ML systems.
They agree to extend the scope and promise an experiment on real-world data in the revised version.

Rev Sm7a appreciates the valuable critique of the considered transformer architectures, with regrets that the authors do not propose another architecture and only provide evidence on synthetic data. Authors state that standard causal masking may offer a reasonable intermediate solution.